# Miracle Stories in Motion—On the Three Editions of *Guangshiyin Yingyanji*

Chon Iat Lai

Department of History, School of Humanities, Tsinghua University, Beijing 100084, China;
ism4c21victor@163.com

**Abstract:** Previous studies have assumed that the purpose of *Yingyanji* was to produce texts that are proselytistic or evangelical. Through the analysis of *Guangshiyin Yingyanji*, we find that lay people have created *Yingyanji* for a long time. Its main purpose was not to spread religion, but to record regional memories and family beliefs, which were mainly circulated among friends and relatives. Moreover, the miracle stories contained in *Guangshiyin Yingyanji* often have different versions within the three systems of *Zhiguai*, *Yingyan*, and *Gantong*. Through an analysis of these different versions, we can better grasp the purpose of rewriting texts under different systems, and the struggle for ideas which they embody.

**Keywords:** *Guangshiyin Yingyanji*; miraculous stories; *Zhiguai*; faith competition; lay people

## 1. Introduction

Previous research on Buddhist miracle stories or "Buddhist auxiliary texts" (釋氏輔教之書)[1] (Lu 1981, p. 54) from the pre-Tang Dynasty can generally be sorted into two approaches: the influence of Buddhism on Six Dynasties literature, and proselytizing Buddhist literature. The first approach emphasizes the influence of Buddhist sutra stories on *Zhiguai* (志怪) and Six Dynasties literature. This research examines the impact of Buddhist themes and concepts on *Zhiguai*[2] while exploring the relationship between "preaching to people and leading them to conversion" (唱導[3]) and "Buddhist auxiliary texts."[4] However, this approach tends to treat "Buddhist auxiliary texts" as indirect research subjects and does not analyze their nature and content in-depth.

The second approach originated with Lu Xun and considers "Buddhist auxiliary texts" as an independent research subject.[5] The definition and scope of "Buddhist auxiliary texts" were further refined in later research conducted by J. Li (1985, pp. 62–68), emphasizing that the authors of these texts aimed to spread Buddhist teachings and doctrines. Q. Zhang (2018, pp. 39–49) believed that the emergence of these works was related to the Buddhist suppression and anti-Buddhist debates of that time, while Q. Zhang (2018, pp. 39–49) and Cao (1992, pp. 26–36) emphasized their relationship to the disputes between body shape and spirit, as well as the disputes between native and foreign cultures, while acknowledging the political intentions shared with other *Zhiguai* works. However, these related inferences were mainly based on circumstantial evidence such as the authors' intentions or historical context, with less discussion on the direct evidence from the works, or differences in the nature of the texts.[6] To further clarify the nature of "Buddhist auxiliary texts", it is necessary to start the discussion with the earliest extant texts of this type: the three editions of *Guangshiyin Yingyanji* (觀世音應驗記).

*Guangshiyin Yingyanji* is a collective term for three different editions: 1. The first edition, known as the *Guangshiyin Yingyanji* (光世音應驗記), was written by Fu Liang 傅亮 (374–426) and is referred to as the *Fu* edition. 2. The second edition, called the *Xu Guangshiyin Yingyanji* 續光世音應驗記, was written by Zhang Yan 張演 in the mid-fifth century, and is referred to as the *Zhang* edition. 3. The third edition, named the *Xi Guanshiyin*

*Yingyanji* 繫觀世音應驗記, was compiled by Lu Gao 陸杲 (459–532) in 501, and is known as the *Lu* edition.

Although these three editions were written by different authors, the later two editions mentioned the early edition and claimed to inherit its subject and compile it. Unfortunately, these three editions were lost in China after the Tang Dynasty, but they were rediscovered at the Shōren-in Temple (青蓮院) in Kyoto during the mid-20th century.

Since the rediscovery of the three different editions of "*Guangshiyin Yingyanji*" in Kyoto, scholars from various countries have conducted research on it. In terms of textual organization, the two annotated editions by Makita Tairyō (1970) and Dong Zhiqiao (2002) are considered the best. The former excels in its historical comparison, while the latter corrects many errors in the original text and provides additional linguistic supplements. In addition, scholars such as Komina (1982, pp. 415–500), X. Zhang (2013, pp. 54–68, 405–17), and C. Sun (1998, pp. 201–28) have conducted research on the circulation and nature of the *Guangshiyin Yingyanji*,[7] or have introduced the belief in Guanyin prevalent during the Six Dynasties period (Makita Tairyo, 1970, pp. 109–56; C. Sun 1998, pp. 201–28; Gu 2015; Xu 2012). However, these studies have mainly focused on organizing the texts, and there is still much work to do regarding the generation of individual stories and their cross-textual transmission. Through an analysis of cross-textual transmission, we can address the following two questions: What is the nature of *Guangshiyin Yingyanji* and the Buddhist auxiliary texts? Also, are there genres of Buddhist auxiliary texts, and what might be their distinctions?

## 2. Writing Miracle Stories—Starting with the Three Editions of *Guangshiyin Yingyanji*

### 2.1. From "Sharing between Like-Minded Individuals"(傳諸同好) to "Extraordinary Worldly Transmission"(神奇世傳): Why There Are Three Editions of Guangshiyin Yingyanji

In previous studies, the three editions of *Guangshiyin Yingyanji* were generally treated as a homogeneous entity. Furthermore, researchers tended to analyze these texts from the standpoint of missionary activities and their function as sermon sources. However, these claims only provide indirect or relatively recent evidence, and often lack any direct evidence regarding the actual purpose of the writing found in its prefaces. While Sun Changwu and Komina Ichirō have recognized the importance of these prefaces and pointed out differences between the early *Fu* and *Zhang* editions, as well as the later Lu Gao edition, specific distinctions and reasons for these distinctions remain unexplained. To address these questions, it is necessary to analyze the three prefaces first. They are listed as follows:

> Fu Liang: Xie Qingxu once wrote a volume of *Guangshiyin Yingyanji* in one roll, consisting of over ten stories, and gave it to my father. I kept it when I resided in Huiji, I lost it while fleeing from the war. Recently, upon returning to this place, I sought it but could not find it anymore. Seven stories I remember clearly, but I cannot recall the rest. Therefore, I have written down what I remember to please like-minded believers.

> 傅亮：謝慶緒往撰《光世音應驗》一卷十餘事，送與先君。余昔居會土，遇兵亂失之。頃還此境，尋求其文，遂不復存。其中七條具識，餘不能復記其事。故以所憶者更為 此記，以悅同信之士云. (Dong Zhiqiao, 2002, p. 1)

> Zhang Yan: Since my youth, I have received teachings and followed the great Dharma, always revering to the supernatural and expressing my admiration. I have long cherished the idea of compiling these records but have not yet accomplished it. When I saw the collection by Fu, it deeply resonated with me. Thus, I decided to write down what I have heard and add it to the end of his text to share it among like-minded individuals 同好.

> 張演：演少因門訓，獲奉大法，每欽服靈異，用兼緬慨。竊懷記拾，久而未就。曾見 傅氏所錄，有契乃心。即撰所聞，繼其篇末，傳諸同好云. (Dong Zhiqiao, 2002, p. 28)

Lu Gao: In the past, an esteemed scholar Xie Qingxu recorded over ten miraculous stories about Guangshiyin and presented them to the Magistrate of Ancheng, Fu Yuan, who was also known as Fu Shuyu. The Fu family resided in Kuaiji, but they lost it during the chaos caused by Sun En. Fu Yuan's son, Fu Liang, who was also known as Fu Jiyou, still remembered seven of those stories and wrote them down. My ancestral uncle, Zhang Yan, who served as an Imperial Secretary, also known as Zhang Jingxuan, separately recorded ten stories to continue Fu's compilation. These seventeen stories have been passed down to the present. Fortunately, I had the opportunity to receive the Buddha's teachings and embraced them since my youth. When I read scriptures describing Guangshiyin, I felt a deep sense of reverence. Additionally, I have seen various contemporary writings and stories that are continuously transmitted by the wise, and their accounts of miraculous events are countless. This has made me realize that the sacred spirits are extremely close, and I am filled with gratitude. I believe that every person's heart has the power to be genuinely moved, and according to the principles of sacred teachings, there must be an inherent force that can be activated. If we can be moved and seek such activation, how can it not have an impact? It is a source of encouragement for virtuous men and virtuous women. Now, in the first year of the Zhongxing reign period of the Southern Qi dynasty(AD 501), I respectfully compiled this volume consisting of sixty-nine stories to connect the works of Fu and Zhang. By arranging them together, readers can see them simultaneously. If there are future wise individuals who continue to hear and learn, they can add to what I have left behind. May this extraordinary worldly transmission widely spread the faith. The details and summaries contained herein are based on what I have heard and know. If you want a detailed examination of it, then we must wait for the insights of other knowledgeable individuals.

陸杲：昔晉高士謝字慶緒記光世音應驗事十有餘條，以與安成太守傅瑗字叔玉。傅家在會稽，經孫恩亂，失之。其子宋尚書令亮字季友猶憶其七條，更追撰為記。杲祖舅太子中舍人張演字景玄又別記十條，以續傅所撰。合十七條，今傳於世。杲幸邀釋迦遺法，幼便信受。見經中說光世音，尤生恭敬。又睹近世書牒及智識永傳，其言威神諸事，蓋不可數。益悟聖靈極近，但自感激。信人人心有能感之誠，聖理謂有必起之力。以能感而求必起，且何緣不如影響也。善男善女人，可不勗哉！今以齊中興元年，敬撰此卷六十九條，以繫傅、張之作。故連之相從，使覽者并見。若來哲續聞，亦即綴我後。神奇世傳，庶廣殞信。此中詳略，皆即所聞知。如其究定，請俟殞識。 (Dong Zhiqiao, 2002, pp. 57–58)

"Like-minded believers" (同信) certainly refers to believers who share the same faith, but "like-minded individuals" (同好) cannot simply be regarded as a friend in the general sense. Here, it specifically refers to a circle of friends with similar interests and intellectual attainment. For example, in Yang Liu Fu (楊柳賦), Kong Zang (孔臧) states: "Thus, friends with shared interests gather, sitting together in groups. Discussing the Dao and drinking wine, flowing rivers, and floating cups." (Fu Yashu, 2011, p. 449) In Zhi Gong Lun (至公論), Cao Yi (曹義) also mentions: "Those who are calm and noble, and share the same interests, are the best of friends." (Cao Yi comp., Yan Kejun ed., 1958, p. 1163) In certain personal works or accounts, they would only circulate within these social circles. For instance, when Cao Zhi (曹植) mentioned his writings in a letter, he said, "Although I have not been able to hide them on famous mountains, I will transmit them among those with those who have shared interests." (Chen Shou, 1982, p. 559) Huiyan (慧嚴) once "complained about the verbosity of the Mahaparinirvana Sutra, so he edited and condensed it into several volumes and copied two or three to share with those with shared interests."[8] Therefore, the sources of and intended audience for the Fu and Zhang editions were limited to scholars or regional communities involved in the same faith. The prefaces do not excessively emphasize their own beliefs but rather highlight the origin of the stories and the desire for recognition from specific Buddhist communities. They were not written for missionary purposes, but rather

designed as booklets for one's own personal social and religious exchanges, intended for internal circulation.[9]

> *This kind of internal communication can be fully exemplified through the interactions and kinship relations among the three editors(Figure 1). The seven stories in the Fu edition*[10] *had originally been given to Fu Liang's father, Fu Yuan (傅瑗), by Xie Fu (謝敷). One of the stories was imparted from his father's friend, Xi Chao (郗超), whose father had recruited Xie Fu. Thus, the Fu family, the Xi family, and the Xie family can be said to have all belonged to the same social circle.*

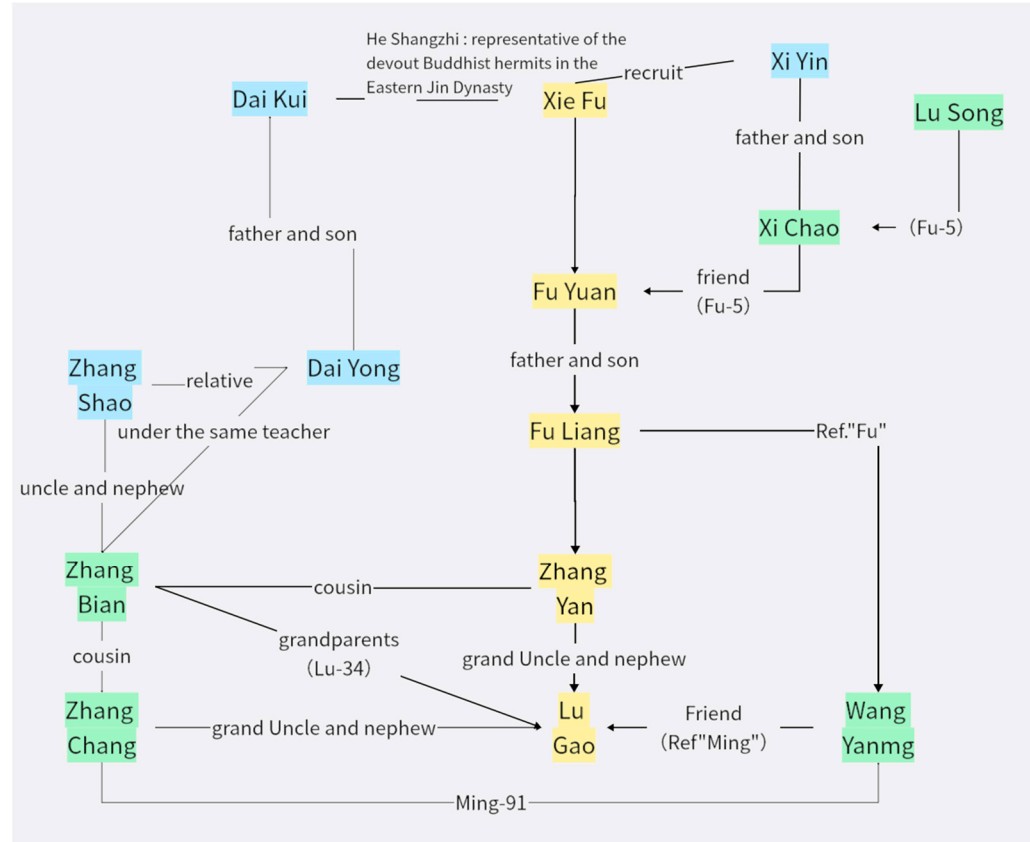

**Figure 1.** Internal communication of three editions of Guangshiyin Yingyanji Yellow: writer or inheritor; Green: story teller; Blue: related parties.

Although the *Zhang* edition was inspired by Fu Liang's writings, it is unclear whether there was any interaction between their families. However, it is evident that the authors of the *Zhang* and *Lu* editions shared the same circle of friends and kinship networks. In the preface, Lu Gao explicitly states that Zhang Yan is his maternal uncle, and also records the affairs of Zhang Yan cousins, who are related to Lu Gao's maternal grandfather, Zhang Chang (張暢). Lu Gao's writing also references Wang Yan's (王琰) *Mingxiangji* (冥祥記), which, in turn, draws from a biography written by Lu Gao's other maternal uncle, Zhang Bian (張辯).[11] This indicates that there is another layer of friendship and kinship between the Zhang, Lu, Dai, and Wang families. Their stories circulated within these circles, becoming shared cultural and intellectual resources among religious communities.

However, there was a change in the case of Lu Gao. In the preface, he emphasizes his family's religious beliefs which he embraced from a young age, as well as his enthusiasm for promoting miracle stories. This is reflected in the selection of sources and the way that the compilation was made. Lu Gao goes beyond the original sources and extensively references knowledge from sources outside his family, such as "recent writings and the wise". Thus, in this case, *Yingyanji* was no longer just a compilation of family memories but in-

stead became a consciously composed collection of missionary stories. The purpose of the writing shifted from an exchange of stories to the act of compiling texts, and its original social and religious roles became reversed. Emphasis was now on affirming personal beliefs. This contrast becomes more apparent when compared to the preface of Wang Yan's contemporary work, *Mingxiangji*, in which Wang provides a more detailed description of his personal journey of faith, emphasizing that the stories serve as tools for proving his beliefs and stating, "If the efficacy of the scriptures is revealed, the intent of the evidence is the same; the events are not different, so we follow the same path." However, the missionary aspect was not prominent in both works until the preface of *Mingbaoji* (冥報記), during the early Tang Dynasty, when the notion of "persuading people" first emerged.

Fu Liang mentions that he inherited the stories written by Xie Fu. Thus, *Yingyanji* had already emerged by the mid-Eastern Jin Dynasty. From Fu Liang's preface, one can see that Xie Fu followed the same approach of "pleasing believers of the same". Hence, *Yingyanji* appeared over one hundred years before Lu Gao and Wang Yan, but there is no evidence of similar practices among monks.[12] During the same period as Lu and Wang, there was a growing number of secularly authored collections of verified stories, such as *Xuanming Yan* (宣明驗) and *Buxu Mingxiangji* (補續冥祥記), as well as the initial biographies of Buddhist monks. Therefore, the nature of verified stories may have been influenced by factors such as the compilation of books in the Qi and Liang dynasties, and the development of Buddhist monastic communities, leading to the emergence of characteristics that facilitated their dissemination among believers.

### 2.2. Regional Memories: Sources for and Composition of the Guangshiyin Yingyanji

This difference in intention becomes more evident in the context of its sources and the writing process, where there is both continuity and discontinuity between them. Komina Ichirō has analyzed the sources of each story in the *Yingyanji* in detail, pointing out that Fu Liang's and Zhang Yan's sources mainly stem from personal accounts, acquaintances, or the narratives of monks. Lu Gao also inherited these methods to some extent (Komina 1982, pp. 418–500). However, Komina Ichirō does not answer an important question: Why do many of its sources originate with monks who intend to propagate Buddhism, while the *Yingyanji* itself does not originate from them?

To answer this question, we must first set aside any assumption of "missionary" activities and examine how the *Yingyanji* was composed. Both Fu Liang and Zhang Yan mention that, in addition to the accounts recorded by predecessors, such as Xie Fu and Fu Liang, they also included stories based on their memories and what they had heard, at times explicitly noting the sources of their information. Lu Gao also frequently mentions different editions of the same story, many of which come from his own family or extended relatives. These records and personal experiences often indicate a story's principal source and background. In Fu Liang's text, there are three stories related to the region of *Kuaiji* (會稽), and Xie Fu himself is from that region, along with Fu Liang and his father residing there; In Zhang Yan's text, there are six stories connected with the region of Jingzhou (荊州), where his father had served as a military advisor and a magistrate in his early years.[13] Lu Gao mostly recorded stories from various places in *Yangzhou* (楊州), which can be attributed to his father's position as an official in Yangzhou. Lu Gao also had extensive experience with his own long-term positions there. Therefore, regardless of whether these stories came from monastic or secular sources, their primary attribute is a form of regional knowledge.

Hence, the *Yingyanji* not only served its purpose of recording strange phenomena as a *Zhiguai*, but also served as a repository of family and regional knowledge. This is well illustrated in one of the stories found in Zhang Yan's text [*Zhang*-7][14]:

Sengrong once joined Shi Tanyi in Jiangling to advise a married couple to uphold the precepts. Later, her husband, implicated by the thieves, escaped. The authorities could only capture his wife and send her to prison. On the way to the prison, she encountered Rong and pleaded for his help. Shi Tanyi responded, "You should engrossing focus on reciting the name of Guanyin Bodhisattva, and

there is no other method." The woman immediately began reciting without interruption. During her imprisonment, one night she dreamed of a monk standing between her shoulders and kicking her with his foot, instructing her to leave. Startled, she woke up and found herself freed from the three wooden restraints. Seeing that the gate was still closed and guarded by several gatekeepers, she thought it was impossible to leave, so she put the restraints back on herself. After a while she fell asleep again and dreamt of someone saying, "Why do not you go? The gate is open." Upon waking up, she passed the guard and walked to the gate, miraculously finding it open. She headed southeast for several miles and was about to reach a village. It was dark and obscure when suddenly she encountered someone, initially feeling alarmed and frightened. At the same time, her husband had been hiding in the grass and wandering during the day, and they asked each other about their well-being. They were indeed the husband and wife. They sought refuge with Shi Tanyi, who hid them in a separate place within the temple. Not long after that, a traveling merchant from their hometown arrived, and Shi Tanyi arranged for them to accompany him and escape successfully.

僧融又嘗與釋曇翼於江陵勸一人夫妻戒，後其人爲劫所引，因遂越走。執婦繫獄。融遇途見之，仍求哀救，對曰：“惟當一心念光世音耳，更無餘術。”婦人便稱念不輟。幽閉經時，後夜夢見沙門立其頸間，以足蹴之令去。婦人驚覺，身貫三木忽自離解。見門猶閉，闇司數重守之。謂無出理，還自穿著。有頃得眠，復夢向人曰：“何以不去？門自開也。”既起，乃越人向門，門開得出。東南行數里，將至民居。時天夜晦冥，忽逢一人，初甚駭懼。時其夫亦依竄草野，晝伏夜行，各相問訊，乃其夫妻也。遂共投翼，翼即藏之寺內別處。無何，其鄉人有遠商者，翼令隨去，竟得免也。 (Dong Zhiqiao, 2002, p. 48)

This story was later recorded in both the *Fayuan Zhulin* and the *Taiping Guangji* with reference to the *Mingxiangji*. The overall framework of the story is the same; however, there is some discrepancy in the details, and additional information is provided (see Table 1). The incident took place during the early Yuanjia period, and the layperson mentioned in the story is named Zhang Xing.

**Table 1.** The differing versions of the Sengrong story.

| | Timeframe | Laymen | Place | Receiving Ordination | Refuge |
|---|---|---|---|---|---|
| *Zhang-7* | no | couple | Jiangling | Precepts; Sengrong, Tanyi | Tanyi |
| *Fayuan Zhulin* | early Yuanjia period | Zhang Xing couple | | eight precepts; Sengrong, Tanyi | Sengyi |
| *Taiping Guangji* | early Yuanjia period | Zhang Xing couple | | eight precepts; Sengrong, Tanyi | Sengyi |
| *Xu Gaoseng Chuan* | Early Liang dynasty | couple | Jiangling | five precepts: Sengrong | change to take refuge in businessmen |

Among the three monks mentioned in the story, the records of Tanyi and Sengyi are the most well-documented. Both of them have biographies in the *Gaoseng Zhuan*. Tanyi was a disciple of *Daoan* (道安) and was sent to Jiangling to establish the Changsha Monastery and lead the monastic community. He passed away in the nineteenth year of the Taiyuan era (AD 394). Sengyi was a disciple of Huiyuan (慧遠) and traveled north to study in Guanzhong (關中). In the thirteenth year of the Yongxi era (AD 417), he established a monastery in Kuaiji and subsequently lived there in seclusion for thirty years. He passed away in the twenty-seventh

year of the Yuanjia era (AD 450). By comparison, very few records on Sengrong exist. Both the *Gaoseng Zhuan* (高僧傳) and the *Mingseng Zhuan* (名僧傳) only mention that he was a monk active in the Lushan area of Jiujing. The *Gaoseng Zhuan* also indicates his ability to subdue demons. According to records, he should have been a monk active during the late Eastern Jin dynasty. In summary, Tanyi was a monk active in Jiangling during the second half of the 4th century; Sengyi was a monk active in Kuaiji during the early Liu-Song dynasty; Sengrong was a monk active in Jiujing during the late Eastern Jin period. Each of them operated in different regions, and Sengyi came later than the other two, indicating no apparent connection between them.

When comparing the *Zhang* edition with *Mingxiangji*, the first issue becomes the timeframe of the story. According to *Zhang*, considering the lower limit mentioned in terms of reign titles and events, this story likely took place toward the end of the Jin dynasty. However, according to *Mingxiangji*, it occurred during the early years of the Liu-Song dynasty.[15] The deceased monk, Tanyi, could not have appeared during the early years of the Liu-Song dynasty; therefore, a contradiction exists between the two editions of the story.

If the content of the *Zhang* edition is assumed to be entirely true, then the additional timeframe and altered characters found in *Mingxiangji* would be incorrect. On the other hand, both versions of the *Fayuan Zhulin* and the *Taiping Guangji* state that Tanyi conferred the precepts, but the monk who received them had changed from Tanyi to Sengyi. Furthermore, both versions depict the layman, Zhang Xing, as the protagonist and transform Tanyi, the monk from Jiangling, into Sengyi, the monk from Kuaiji. Consequently, a story about a monk named Tanyi in Jiangling during the late Eastern Jin period saving a couple is transformed into a story about a layman in the early Liu-Song dynasty being rescued from distress.

In addition to the possibility of *Mingxiangji* being modified, one can find potential alterations to another story involving Sengrong in the *Zhang* edition (*Zhang*-6):

> The monk Shi Sengrong was devoted and compassionate. He advised a family in Jiangling to embrace Buddhism and practice it together. Initially, there were several temples dedicated to gods, which were provided for the support of the monks. Sengrong decided to demolish and remove all the pagan temples associated with the laymen's family, so he stayed there for seven days for the Buddhist assembly. After Sengrong returns to this temple, the homeowner of that family suddenly sees a ghost holding a red rope, intending to bind him. The mother became greatly worried and immediately invited a Buddhist monk to chant scriptures, causing the ghost to vanish on its own. Sengrong later returned to Mount Lushan and stayed overnight at an inn along the way. It was raining and snowing that night, and he only fell asleep in the middle of the night. Suddenly, he saw numerous ghostly soldiers, among them a particularly large one wearing armor and carrying a weapon. He sat on the big bed that someone was holding up. The great ghost suddenly exclaimed with a stern voice, "How dare you say that ghosts cannot fulfill other people's wishes!". They attempted to drag Sengrong to the ground. However, before they could act, Sengrong concentrated and chanted the name of Bodhisattva Guanyin. Before his voice faded, a figure resembling a general, over a *zhang* (seven feet) tall, emerged from behind the bed where Sengrong was staying. This figure wore yellow-dyed leather trousers and held a golden disc, confronting the ghost. The ghost was immediately frightened and scattered, and the ghost soldiers in armor were suddenly shattered into pieces.

> 僧人釋僧融，篤志泛愛，勸江陵一家，令合門奉佛。其先有神寺數間，以與之，充 給僧用。融便毀撤，大小悉取，因留設福七日。還寺之後，主人忽見一鬼，持赤索，欲縛之。母甚憂懅，乃便請沙門轉經，鬼怪遂自無。融後還廬山，道中獨宿逆旅。時天雨雪，中夜始眠。忽見鬼兵甚眾，其一大者帶甲挾刃，形甚壯偉，有舉胡床者，大鬼對己前據之。乃揚聲厲色曰：君何謂鬼神無靈耶？便使曳融下地。左右未及加手，融意大不懌，稱念光世音，聲未及絕，即見所住床後，有一

狀若將帥者，可長丈餘，著黃染皮袴褶，手提金枚以擬鬼，鬼便驚懼散走，甲冑之卒然粉碎。 (Dong Zhiqiao, 2002, p. 44)

This story can be divided into two parts: Sengrong's solicitation in Jiangling and his encounter with ghosts in Mount Lu. Sengrong had already left after preaching in Jiangling, and it was other monks who managed to exorcise the ghosts that disturb the households of believers. This appears to be unrelated to Sengrong's later encounter with ghosts. (*Zhang*-6) clearly describes that both Tanyi and Sengrong were involved in the proselyting in Jiangling, and that the woman sought help from Sengrong but ended up taking refuge in the temple where Tanyi resided. (*Zhang*-7) depicts Sengrong's involvement in the solicitation in Jiangling, yet it was the other monks who resolved the ghost encounter for the host, while Sengrong encountered ghosts during his solitary training in Mount Lu. Both stories begin with Sengrong's involvement in proselyting, but only in a certain portion of the stories. Therefore, there is a possibility of later recompilation.

When comparing the records of the three monks in the Buddhist biographies, Monk Yi does not have any miraculous incidents associated with him. The miraculous incidents attributed to Monk Tanyi are all related to relics and Buddha statues, emphasizing his sincere faith rather than his inherent supernatural abilities. Only Sengrong is described as having the ability to subdue ghosts and spirits through his austere practice. Therefore, we can suggest that Sengrong is likely to have been added as a character with supernatural abilities to the story originally centered around Tanyi, who had been the main protagonist in Jiangling. Sengrong's encounter with ghosts was an additional story placed in the background of Jiangling. Both stories involving Sengrong were combined and included in the *Xu Gaoseng Zhuan* (續高僧傳), where Sengrong was reimagined as a monk with supernatural abilities during the early Liang dynasty. Although both stories are set in Jiangling, the presence of the other monks has been removed, and the act of seeking refuge alone was altered to seeking refuge together with a merchant, making Sengrong the sole protagonist of the story. The *Xu Gaoseng Zhuan* explicitly states that he came from the Donglin Temple in Jiujian, which indicates a possible reference to another version of the story circulating in Jiujian.[16]

Apart from the story itself, the construction of regional knowledge can be observed through its narrators. This characteristic is particularly evident when comparing stories told by northern immigrants. The stories of those who migrated from the north to the south were not recorded by the author, but rather passed down by familiar monks or laymen.[17] Therefore, they were based upon certain points during the war, had relatively simple storylines, and did not record the subsequent experiences of any individuals involved, or other versions of the story. The other stories, however, were narrated by the individuals involved or their relatives and friends, because their protagonists had lived in the Southern Dynasties. Some stories even have multiple narrators, resulting in more complex narratives and multiple versions.

This characteristic of multiple narratives can be illustrated through the story of the "Pengcheng widow" (*Lu*-63).

The Pengcheng widow came from a family devoted to Buddhism, and she was diligent in her practice. She had lost all her relatives, leaving only one son who listened to her teachings. The son was extremely filial, and the bond between the mother and son was filled with love and compassion. In the seventh year of the Yuanjia era (AD 430), her son accompanied Dao Yanzhi on a military campaign against the nomads. The elderly widow bid farewell with tears, repeatedly advising and admonishing her son to observe the precepts and have faith in Guanyin Bodhisattva. The elderly widow was extremely poor, she had nothing to offer the Buddhist assembly, so she often sat in front of the Guanyin statue, lighting a lamp to pray for blessings. Her son was captured by the Wei Kingdom army while carrying out his mission to capture prisoners. Fearing that he might escape, the Wei Kingdom army escorted him to the northernmost border. When the army returned, her son did not come back. However, she kept lighting a

lamp in front of the statue, praying for Guanyin Bodhisattva's help. During the same period, her son also prayed day and night in the north. One night, he suddenly saw a light shining brightly at a distance of a hundred paces. He tried to approach it, but the light disappeared. Then, he saw it again in front of him, as if beckoning him. He thought it was a divine phenomenon, so he followed the light. After every sunset, the light would be illuminated again. Therefore, he stopped in a village to beg for food during the day and continued his journey at night guided by the light. He traversed mountains and valleys as if they were level ground, traveling thousands of miles until he returned to his hometown. Upon his arrival, he saw his mother still kneeling in front of the lamp, her face illuminated by its light. He realized that the light he had seen before was the lamp before the statue. The news spread far and wide, and everyone rejoiced in their miraculous experience. The mother and son redoubled their efforts in their spiritual practice. After his mother's passing, the son decided to become a monk. Later, he sought a master and disappeared; no one knew where he was.

Another version tells of the widow. After she lost her son, she constantly lit a lamp in front of the Guanyin statue and recited the Guanyin Sutra day and night, hoping to have a vision of Guanyin. However, she is also afraid that her son may have already perished. She also performed seasonal ancestral rituals. The nomads treated her son as a slave and assigned him to herd the animals. Every time during the ancestral ritual, her son would dream of returning to partake in the offerings. After the widow's sincere devotion for over a year. One day, while her son was in the mountains, he suddenly saw a pillar-like light, approximately ten steps away, that quickly moved beyond his reach. He pursued it persistently and finally returned home after ten days. Upon his return home, he witnessed the light leading directly to the Guanyin statue, while his mother was prostrated in front of it.

There are two versions of this story. I copied it from the *Xuanyanji* by Gao Chao. I showed them to the provincial official He Yi of Nanyuzhou. He Yi, known as a diligent and honest scholar, said, "I have heard this story since my childhood. The widow was my maternal grandmother. I have often heard my family reiterate her tale, saying that she tore a lot for her lost son. Her tears fell on the lamp, causing it to burst. Her cheeks were scalded and burned by the lamp oil.

彭城嫗者，家世事佛，嫗唯精進。親屬並亡，唯有一子，素能教訓。兒甚有孝敬，母子慈愛，大至無倫。元嘉七年，兒隨到彥之伐虜。嫗衛弟追送，唯屬弁歸依觀世音。家本極貧，無以設福，母但常在觀世音像前然燈乞願。兒於軍中出取獲，為虜所得。慮其叛亡，遂遠送北堺。及到軍復還，而嫗子不反，唯歸心燈像，猶欲一望感淚。兒在北亦恆長在念，日夜積心。後夜，忽見一燈，顯其百步。試往觀之，至徑失去。因即更見在前，已復如向，疑是神異，為自走逐。日沒，還復見燈，遂晝停村乞食，夜乘燈去。經歷山險，怳若行平。輾轉數千里，遂還鄉。初至，正見母在像前，伏燈火下。因悟前所見燈即是像前燈也。遠近聞之，無不助為憙。其母子遭荷神力，倍精進。兒終卒供養，乃出家學道。後遂尋師遠遁，不知所終。

一說嫗既失子，恆燃燈觀世音像前，晝夜誦觀世音經，希感聖神，望一相見，又恐或已亡沒，兼四時祠之。虜以嫗子為奴，放牧草澤。母祠之日，輒夢還饗。母積誠一年，晝夜至到。後兒在山中，忽見一光如柱形，長一丈，去已十步，而疾走不及。逐之不已，得十日至家。至家，見光直歸像前，母正稽顙在地。

有二本如此云。呆抄《宣驗記》，得此事，以示南豫州別駕何意。意，篤學厚士也。語呆：此嫗，其外氏。固從已小時數聞家中叙其事，云嫗失兒，恆沾淚，淚下燈爆，雨頰遂爛，其苦至如此。(Dong Zhiqiao, 2002, pp. 194–95)

In addition to He Yi's version, there are three versions of this story. The general idea of the story revolves around a mother and son from Pengcheng. The son was captured by

the enemy during the Northern Expedition in the seventh year of the Yuanjia era (AD 430), but later returned. Throughout his journey south, he was guided by a lamp, and upon reaching home, he discovered that his mother had been praying with a lamp in front of the statue all along.

Interestingly, *Xuanyanji* (宣驗記) also records another strikingly similar story, which is cited in *Bianzheng Lun* (辯正論), *Shishi Liutie* (釋氏六帖), and *Taiping Guangji*. The three citations are essentially the same. This is the version cited from *Bianzheng Lun*:

> "The story of Che's mother lighting the lamp to pray, and her son unexpectedly coming back": The story of Che's mother is about her son suffering during the "Qingni Incident" caused by the King Luling of Song, which was captured by Fofo caitiffs and imprisoned in the enemy barracks. His mother has always been a Buddhist, so she immediately started to light seven lamps in front of the Buddha statue. She wept earnestly day and night, praying for her son's liberation. This went on for years. Suddenly, her son managed to escape and return, traveling alone on foot for seven days. He lost direction due to the cloudy weather, and he saw seven segments of firelight in the distance and ran toward them. It appeared to be a village, so he intended to seek refuge, but he was unable to reach it continuously. In addition, after seven nights, he unknowingly arrived home. He saw his mother still praying in front of the Buddha and lit seven lamps. At that moment, they both realized the power of the Buddha. From then on, they devoted themselves to practicing acts of charity and endurance.

> 車母燃燈不期兒至。車母者，遭宋盧陵王青泥之難為佛佛虜所得，在賊營中。其母先來奉佛，即燃七燈於佛前。晝夜精心哭觀世音，願子得脫。如是經年，其子忽得叛還。七日七夜行獨自南走，值天陰不知西東。遙見有七段火光，望火而走。似村欲投，終不可至。如是七夕，不覺到家。見其母猶在佛前伏地，又見七燈，因乃發悟。母子共 談知是佛力，自後懇到專行檀忍。[18]

Here, the timeframe of the story changes to the Qingni Incident in the 14th year of the Yihe era (AD 418), referring to Liu Yizhen's (劉義真) retreat from Guanzhong (關中). Although there are differences between the two accounts in terms of the names of the individuals involved, the objects of offering, and the details about lighting the lamps,[19] the theme and structure of the story are indeed similar, in which a mother offers lamps to the Buddha and prays for her son's return. There are two possibilities here: one is that *Guangshiyin Yingyanji* recorded two highly similar stories; the other is that they draw on one another. Regardless of the outcome, it can be inferred that this particular story was widely popular when *Guangshiyin Yingyanji* was written, leading to the emergence of several versions. Whether it was rewritten, or just a selection of a particular version, *Lu* abandoned the story of a son being saved through the mother's offering to the Buddha, and instead chose (or composed) a version that emphasized the family's devotion to the Guanyin, resulting in his salvation.

If we focus solely on determining which version is true, or if there is a relationship between copying and rewriting, we may overlook the unique qualities of *Yingyanji*. Only by combining local knowledge with the establishment of familial and regional characteristics can one discover the intertextuality among different stories, as well as the complex interactive relationships between them.[20] When Liu Yizhen compiled *Xuanyanji*, only twenty years had passed since the Qingni Incident. In contrast, Lu Gao, who recorded the *Lu* edition, was separated from Dao Yanzhi's Northern Expedition by seventy years, and from the Qingni Incident by nearly a hundred. It would have been difficult for him to determine the exact timing of these events. Additionally, the story would have been spread over different regions and through different battles. In addition to the possibility that the existing version of *Xuanyanji* differs from what Lu Gao saw, it is also possible that Lu Gao supplemented the story based on other versions he had heard, or other augmented versions he had copied. Lu Gao said that "the story was widely known, and everyone enthusiastically supported it". We can imagine that a story of successful escape and return home must have

been rare and deeply impactful. Therefore, the focus is not on determining which version is true, as *Xuanyanji* provides only the earliest existing version. The significance lies in how the story of the "Pengcheng widow" carries the collective memory of repeated failure as well as the loss of family and loved ones during the Northern Expeditions in the Jin-Song transition period. In this way, this original narrative has had quite a lasting impact.[21]

Its multiple narratives make it difficult to ascertain the truth of the story, but the story itself carries specific collective memories, allowing us to glimpse into how the story was constructed, as seen in (*Lu*-32):

> Zhu Lingshi, a native of Pei, was a meritorious minister of Emperor Gaozu of the Liu-Song dynasty. In the early period of the Jin Dynasty's Yixi era, he served as the magistrate of Wuxing Wukang. At that time, there were many wicked people in the county, and Lingshi executed and killed a large number of them, exceeding the proper limit, which could have led to his death sentence. The court ordered Zhang Chongzhi to investigate the matter, and Lingshi was arrested and imprisoned, awaiting execution. The family filed a lawsuit at the time, but a final verdict has not yet been reached. Corrected: At that time, there was a monk named Shi Huinan who was an old acquaintance of Lingshi. Someone informed Shi Huinan about the news, and he went to visit Lingshi in prison. He taught Lingshi to recite the name of Guanyin and also left a statue of Guanyin for worship. Lingshi was already a believer in Buddhism, and now that he was facing adversity, he became even more devoted to his worship, continually reciting the name of Guanyin. After seven days, his shackles were miraculously unlocked. The prison guards were amazed, so they reported it to Zhang Chongzhi. Zhang suspected that Lingshi became thin during the period. They try to put the shackles back on, but they did not fit. They still believed it was just a coincidence, so they tightened the shackles again. However, after a few days, the shackles loosened again. This situation happened three times, so Zhang Chongzhi reported this miraculous incident. While detailed discussions on the matter had already refuted the accusations against Lingshi, Zhang Chongzhi's report also arrived, so they immediately released Lingshi and resumed his post. Both Lingshi and his brothers achieved great success.

> 朱齡石，沛人也，為宋高祖功臣。晉義熙初，作吳興武康令，時縣有兇猾，齡石誅殺過多，當死。朝廷使張崇之檢校其事，被收錄，繫在獄中，當死。家人訟訴，是非未辯。時有道人釋惠難與石有舊，乃往告，入獄看之。因教其念觀世音，又留一人像與供養。齡石本事佛，並窮厄意專，遂一心係念。得七日，即鎖械自脫。獄吏驚怪，以故白崇。崇疑是愁苦形瘦，故鎖械得脫。試使還著，永不復入。猶謂偶爾，更釘著之。又經少日，已得如前。凡三過，崇即啓以為異。爾時都下前論詳其事，已破申。會崇至，還復縣，齡石亦終能至到，兄弟有功名。　　(Dong Zhiqiao, 2002, p. 124)

According to the "Biography of Zhu Lingshi" in the *Songshu* (宋書), Zhu Lingshi's indiscriminate killings occurred after the Jin Dynasty invaded Shu in the tenth year of the Yixi reign (AD 414). It states, "Initially, Lingshi pacified the rebellion in the Shu region, and the number of people he executed was limited to the rebel leader's clan. However, Hou Chande rebelled and many people were implicated and executed". This shows that the incident had a significant impact at that time. However, this incident does not correspond to the account mentioned here. Nevertheless, years before the invasion of Shu, Zhu Lingshi was appointed as the magistrate of Wukang and was indeed involved in the execution of local ruffians in Wuxing. During his tenure, he lured and killed dozens of bandits, bringing peace to the county. However, there is no record of him being held accountable, and he was soon promoted.[22] Therefore, it can be inferred that the stories in *Lu* should be combined with these two incidents. Additionally, similar narratives can be found in other anecdotal stories about Zhu Lingshi.[23]



Accordingly, it can likely be concluded that the stories in *Yingyanji* may not have been recorded accurately, but instead draw upon certain real events and bear certain collective memories. Such ambiguity is not uncommon. When Baochang (寶唱) collected stories about Shanmiaoni's (善妙尼) self-immolation, which happened several decades prior, there were already three different versions circulating during the 17th Yuanjia (AD 440), Xiaojian (AD 454–456), and Daming (AD 457–464) periods.[24] Therefore, the fluid nature of legends should not be underestimated. Correspondingly, these collective memories serve as the prototypes of stories which are constantly rewritten and transmitted, allowing them to continue to be passed down.

In summary, the reason why the *Yingyanji* was not compiled by monks is precisely that it was a compilation of stories put together by an author who had either heard or experienced them, and then shared them with their regional community. Therefore, Xie Fu, who resided in Mount Kuaiji, passed on the stories to the Fu family, who in turn added on the local stories they had heard in Kuaiji. Zhang Yan, a relative of the Fu family, obtained the version from Fu Liang and added stories that were circulated in Jingzhou. Lu Gao, building upon the work of his predecessors, incorporated many local stories from Yangzhou and had interactions with Wang Yan and He Yi.[25] They continually supplemented the stories told by their relatives and friends based on the foundation of previous works, resulting in a collection of stories circulated within a specific group. It is precisely due to the nature of this internal circulation that stories often do not require an accurate time. This is particularly evident in Zhang Yan's edition, where out of ten stories, only three indicate some time of occurrence.[26] Among those seven without dates or time markers, four had dates added in later records and some details were added, too, entering into the realm of biographical records.[27]

This tendency gradually disappears in Lu Gao's version, and other works with a heavier emphasis on proselytizing, such as the *Mingxiangji*. They require explicit time markers and merits to enhance credibility and incorporate various accounts and arguments to strengthen their persuasiveness.[28] At the same time, their sources of material continued to expand. Lu Gao collected stories from various regions and various textual sources to demonstrate the universality of Guangshiyin worship. On the other hand, the *Mingxiangji* aimed to collect all miracle stories since Emperor Ming's dream of Buddha, focusing on the spread of Buddhism to the East and the development of Han Chinese Buddhism.[29] Both would gradually move away from the local, inward-facing approach to compilation, instead shifting their focus from the family to the broader society, and expanded the scope of their object of worship from specific religious figures to the entire Buddhist tradition.

Miracle stories are not independent products of a specific time and place, but are constructed through the layering of first-hand witnesses, narrators, and recorders. These stories also generate different versions in different regions. Even after being recorded, each version still contains differences based on factors such as region and perspective in different types of texts. Therefore, an analysis on the dissemination of miracle stories cannot be limited to the *Yingyanji* itself; it is necessary to examine the overall process of their dissemination. Based on different writing perspectives and factors, these versions can be classified into different textual systems. The next chapter will focus on the study of the rewriting and generation of supernatural stories within different textual systems, examining the stories within the context of their overall transmission process. By exploring the continuous compilation and rewriting of supernatural stories, as well as the writing characteristics of different textual systems, we can better understand the fluidity of supernatural stories.

## 3. *Zhiguai*, *Yingyan* (應驗), and *Gantong* (感通): The Compilation of Miracle Stories

Since the nature of *Yingyanji* differs from later "Buddhist auxiliary texts", discrepancies within the same story among different dissemination systems become significant. We may classify them based on the *Zhiguai* system, which has no Buddhist leanings; the *Yingyan*[30] system, which was written by Buddhist laypeople; and the *Gantong*[31] system written by Buddhist monks. Unlike the later-developed chuanqi, Yingyan, and Gantong

in Zhiguai, they still retain descriptive traits and are mostly of short length. However, they exhibit different emphases when narrating the stories, leading to their selection and the rewriting of the stories. As a result, the same story may take on various appearances and versions in the records of the three distinct narrative systems. Below, we will compare Yingyan with Zhiguai first.

*3.1. Zhiguai and Yingyan: Analysis of Stories with Parallel Non-Religious Literature*

Of the six stories with parallel texts in nonreligious literature, five of them have pre-Tang texts. Here, detailed discussions will be conducted on four of them.[32]

*Fu*-7: Monk Zhu Fayi
Direct source: Fayi told Fu Liang's father when they traveled together
Parallel texts: *Mingseng Zhuan, Gaoseng Zhuan, Fayuan Zhulin* quoted *Mingxiangji, Fayuan Zhulin* quoted *Shuyiji* (述異記), *Taiping Guangji* quoted *Shuyiji, Bei Shan Lu* (北山錄)

This story does not originate from Xie Fu, but from Fu Liang, who inherited it from his father. Therefore, it should be considered the original version of the story. The story's essence is that the monk Zhu Fuyi became ill and turned to Guanyin for salvation. Later, he dreamed of a monk cleansing his intestines, and upon waking up, he miraculously recovered. The story later appeared in *Shuyiji*,[33] *Mingxiangji*, and *Mingseng Zhuan* during the Qi and Liang dynasties. It continued to be disseminated in the religious literature, such as the *Gaoseng Zhuan* and the *Fayuan Zhulin*.

Makita Tairyō argues that two similar stories found in the *Fayuan Zhulin* come from *Mingxiangji* and the *Fu* edition. He considers the attribution of the story to *Shuyiji* a mistake, which was subsequently inherited by the *Taiping Guangji* [Makita Tairyō, 1970, p. 82]. However, this is a simple deduction that simplifies the transmission of the story. Since the *Fu* edition was heard by the author's father and is likely the original record of the story, it serves as a common source for both *Shuyiji* and *Mingxiangji*, with the two being over one hundred years apart in their respective narrations. Therefore, it is rather reasonable for *Fayuan Zhulin* to have adopted the later *Shuyiji* and this hypothesis can be supported by comparing the two versions of *Mingxiangji* and *Shuyiji*.

The two versions recorded in *Fayuan Zhulin* are as follows. *Fayuan Zhulin* quoted *Mingxiangji*:

> During the Jin Dynasty, there was a monk named Zhu Fuyi in Shining Mountain. He was very knowledgeable and particularly adept at Lotus Sutra. He had more than a hundred disciples. In the second year of the Xian'an era (AD 372), he suddenly fell ill and felt strong discomfort in his heart. He always kept praying to Guanyin. One night, he dreamed of a person opening his abdomen and washing his intestines. When he woke up, his illness miraculously disappeared. Fu Liang often said that his father and Zhu Fuyi had a close relationship; every time he mentioned the miracles of Guanyin, he would show great respect.

> 晉始寧山有竺法義。晉興寧中沙門，游刃眾典尤善法華，受業弟子常有百餘。至咸安二年，忽感心氣疾病，常存念觀世音。乃夢見一人破腹洗腸，寤便病愈。傅亮每云：吾先君與義公游處無間，說觀世音神異，莫不大小肅然。[34]

The narrative of *Mingxiangji*[35] and the *Fu* edition is generally the same, but the former adds the lively detail of "during the Xingning period." This account aligns with the record in *Shamen Tanzongsiji* (沙門曇宗寺記) from *Gaoseng Zhuan*,[36] mentioning that Zhu Fuyi "first resided in Baoshan during the Xingning period." *Fayuan Zhulin* cites *Shuyiji* as follows:

> During the Jin dynasty, the Śramana Zhu Fuyi resided in the mountains and was a diligent scholar. He lived in Baoshan in Xingning. Later, he fell ill for a long time, and despite extensive medical treatment, his condition did not improve. He became bedridden and gave up treatment, relying solely on devotion to Guanyin. This continued for several days until one day, while he was sleeping during the

day, he dreamt of a divine being who came to attend to his illness. The divine underwent a surgical procedure, removing and cleansing his intestines and stomach, discovering numerous impurities and cleansing them before putting them back inside. The divine being said, "Your illness has been eliminated." Upon waking up, all his ailments disappeared, and he returned to his normal state of health. According to the scripture (*Lotus Sutra*), Guanyin may manifest as a monk or a Brahmin, which Zhu Fuyi interpreted as the divine being in his dream. Zhu Fuyi passed away in the seventh year of the Taiyuan era (372). Correct: Of the six incidents involving Zhu Zhangshu to Zhu Faye, all were written by Fu Liang, the Prime Minister of the Liu-Song dynasty. Fu Liang stated that his father had interacted with Zhu Fuyi. Whenever Zhu Fuyi recounted these events, his father would feel more respect.

晉沙門竺法義，山居好學，住在始寧保山。後得病積時。攻治備至而了不損。日就綿篤，遂不復自治，唯歸誠觀世音。如此數日，晝眠夢見一道人來候其病。因為治之，刳出腸胃，湔洗腑藏。見有結聚不淨物甚多，洗濯畢還內之，語義曰：「汝病已除。」眠覺眾患豁然，尋得復常。案其經云，或現沙門梵志之像，意者義公所夢其是乎。義以太元七年亡。自竺長舒至義六事，並宋尚書令傅亮所撰。亮自云：其先君與義游處。義每說其事，輒懍然增肅焉。[37]

From the concluding sections of both texts, it can be seen that *Shuyiji* and *Mingxiangji* share a common source. Therefore, it is not possible to affirm *Mingxiangji* while negating *Shuyiji*. It cannot be ruled out that one version was copied by *Fayuan Zhulin* from *Shuyiji*. For example, the mention of the year of death in *Shuyiji* is not found in other texts.[38] Correspondingly, *Shamen Tanzongsiji* does not mix the accounts of *Zhu Fuyi's* studies and the construction of the Jianxin Pavilion Temple into this system. Moreover, from its table of contents, it can be inferred that these two versions have different emphases. Based on this, we can outline the context of the story's evolution:

    1. *Fu* edition --> *Mingxiangji + Shamen Tanzongsiji* --> *Mingseng Zhuan, Gaoseng Zhuan* (+ unknown text), *Fayuan Zhulin*

    2. *Fu* edition --> *Shuyiji* --> *Fayuan Zhulin, Taiping Guangji*

*Lu*-15: Gao Xun

Direct sources: *Shuzhengji* (述征記) and *Zhenyiji* (甄異記)[39]
Parallel texts: *Guanyin Yishu* (觀音義疏) quoting *Yingyanji*, *Bianzheng Lun* quoting *Xuanyanji* and *Xu Soushenji* (續搜神記), *Fahua jing Wenju Fuzheng Ji* (法華經文句輔正記), *Taiping Guangji* quoting *Xuanyanji*, *Sanbao Ganying Yao Luelu* (三寶感應要略錄) quoting *Xuanyanji*

At the end of the text, Lu Gao quotes *Shuzhengji* and *Zhenyiji*, but it is only for reference and not a direct copy. The main body of the story still comes from *Yingyanji*. Both *Xuanyanji* and Lu Gao deliberately omit the variant of selling one's wife, indicating their conscious removal of this detail. Interestingly, in later works such as *Guanyin Yishu* and *Fahua jing Wenju Fuzheng Ji*, there is no avoidance of such detail.

Furthermore, in the version of *Xuanyanji* quoted in *Bianzheng Lun*, the object of the protagonist's prayers is the "Buddha deity", and the divine power of the Buddha deity is emphasized repeatedly, but Guanyin did not appear. However, in the later versions of *Xuanyanji* quoted in *Taiping Guangji* and *Sanbao Ganying Yao Luelu*, both the Buddha deity and Guanyin interchangeably appear in the story, forming the concepts of "reciting Guanyin together" and "devoting oneself wholeheartedly to Guanyin", but also seeking mercy and assistance from Buddha. These reflect the influence of *Yingyanji* on later versions.

*Lu*-34: *Zhang Huoji Shijun*
Direct source: Zhang Chang, the maternal grandfather of Lu Gao
Parallel texts: *Guanyin Yishu* quoting *Yingyanji*, *Taiping Guangji* quoting Yang Jie's *Tansou* (談藪)

Since this story originates from the personal record of Lu Gao's maternal grandfather, Zhang Chang, the events in the story are particularly detailed, including his official

career experience. The story can be divided into two parts: Qiao Wang intends to kill Zhang Chang due to his admonishments, but whenever he has ill intentions, he dreams of Guanyin at night and refrains from inflicting harm. Zhang Chang is later imprisoned for his involvement in Qiao Wang's affairs, so he recites scriptures a thousand times, causing his shackles to break, and is eventually released.

Compared with Lu Gao's account, *Tansou* removes many details such as the religious background and Zhang Chang's admonishment of Qiao Wang, retaining only the records of Guanyin's manifestation twice. Furthermore, the phrase "whenever he has ill intentions and dreams of Guanyin at night" is changed to "when he intends to harm and dreams of Guanyin at night", simplifying the narrative of consistent manifestations to a dream of a divine being.

*Lu*-35: Zhang Da
Direct source: *Zhang Shi Bie Zhuan* (張氏別傳)
Parallel texts: *Bianzheng Lun* quoting *Zhang Shi Bie Zhuan*, *Taiping Guangji* quoting *Zhang Shi Zhuan* (張氏傳), *Shishi Liutie*

This story consists of only about thirty words, describing Zhang Da's imprisonment and subsequent salvation through reciting scriptures, followed by his becoming a monk. However, *Bianzheng Lun* and *Taiping Guangji* only mention that Zhang Da observed a life-long vegetarian diet and abstained from worldly desires, but do not mention him becoming a monk. Furthermore, both sources describe him devoting himself wholeheartedly to meditation,[40] rather than reciting scriptures.

Looking at the differences observed in the versions above, several characteristics can be summarized. First, any overlap between *Yingyanji* and its nonreligious counterpart gradually decreases. There are three instances in Fu Liang's work, one in Zhang's, and two in Lu Gao's. Considering the proportional factor, the later *Yingyanji* and nonreligious literature have less overlap, reflecting a differentiation between *Zhiguai* and *Yingyan* narratives. Second, there is mutual rewriting among the *Zhiguai* and *Yingyan* narrative systems. The *Yingyan* system, influenced by popular religious evangelism, removes descriptions that go against societal ethics, such as "selling wives and children", while such descriptions are preserved in the *Zhiguai* and monk-oriented *Gantong* stories.[41]

Last, the *Zhiguai* narratives are solely concerned with extraordinary phenomena and are not interested in the details of Buddhist beliefs. Therefore, they tend to reduce vivid and intricate stories of spiritual experience into dreams of the divine beings. In response, the *Yingyan* narratives also make modifications to conform them with their own religious beliefs. For example, Chan meditation is changed to scripture recitation, and Buddhist deities are replaced with Guanyin Bodhisattva. Additionally, *Guanghiyin Yingyanji* places more emphasis on the power of Guanyin Bodhisattva compared with other *Yingyan* narratives which are closer to supernatural tales. Apart from changing the object of supplication, stories that cannot demonstrate the power of the spiritual are also removed. Lu Gao selectively removed certain stories several times because "this matter does not reach the level". Traces of these stories can still be found in three other texts:

> In the *Bianzheng Lun*, the *Xuanyanji* is cited, stating that Yu Wen braved the raging waves without fear. When Yu Wen carried salt in Nanhai and encountered strong winds, he silently recited Guanyin's name, and the wind subsided and the waves calmed down. Finally, he got safe.
>
> (This account is also mentioned in *Shishi Liutie*.)
>
> 《辯正論》引《宣驗記》：俞文汎海不畏洪波。俞文載鹽於南海值風。默念觀音，風停浪靜。於是獲安。
>
> (《義楚六帖》亦載)[42]
>
> In the *Fayuan Zhulin*, the *Mingxiangji* is cited, recounting the story of Gu Mai, a resident of Wu County. He was a devout practitioner of Buddhism and served as a military official. In the nineteenth year of the Yuanjia era (AD 443), he re-

turned to Guangling from the capital. When the boat set sail from Shi Tou Cheng, it encountered a headwind, which was an unusual occurrence of strong winds. Despite the ongoing strong winds, the boatmen were eager to move forward. As they reached the middle of the river, the wind and waves grew even stronger, making the situation extremely helpless. He recited the Guanyin Sutra repeatedly, the wind subsided, and the waves diminished. Moreover, a mysterious fragrance permeated the area. Gu Mai was filled with joy and continued to recite the sutra, and thus he reached safety.

《法苑珠林》引《冥祥記》：宋顧邁，吳郡人也，奉法甚謹。為衛府行參軍。元嘉十九年。亦自都還廣陵。發石頭城便逆湖朔，風至橫決。風勢未弭，而舟人務進。既至中江波浪方壯。邁單船孤征憂危無計。誦觀世音經得十許遍。風勢漸歇浪亦稍小。既而中流屢聞奇香芬馥不歇。邁心獨嘉。故歸誦不輟。遂以安濟。[43]

In another account from the *Fayuan Zhulin*, the *Mingxiangji* is cited, narrating the story of Bian Yuezhi 卞悅之, a layman from Jiyin. He resided in Chaogou and was in his fifties without any children. To seek an heir, his wife took a concubine, but she still could not conceive. Desperate for an offspring, Bian Yuezhi recited the Guanyin Sutra a thousand times. Miraculously, after completing the recitation, his concubine became pregnant and gave birth to a son. This incident was recorded in the eighteenth year of the Yuanjia era (AD 442), and the child was already five years old.

(This account is also mentioned in *Taiping Guangji*, but it states it was recorded in the fourteenth year (AD 438) of the Yuanjia era.)

《法苑珠林》引《冥祥記》：宋居士卞悅之。濟陰人也。作朝請居在潮溝。行年五十未有子息。婦為取妾。復積載不孕。將祈求繼嗣。千遍轉觀世音經。其數垂竟妾便有娠。遂生一男。元嘉十八年已五歲 （《太平廣記》亦載，但作元嘉十四年)[44]

Indeed, these stories are rather simple and focused on factual records. Their religious elements are relatively mild and do not emphasize the miraculous nature of the outcomes. Therefore, Lu Gao deemed them to be unfit, as they were not extraordinary, and they were not included.[45] Though *Yingyanji* is often portrayed as being disregarded, it demonstrates Lu Gao's deliberate selection and modification of stories. This indicates a reciprocal relationship between the *Zhiguai* and *Yingyan* genres of mutual rewriting.

### 3.2. The Systems of Xuanyanji and Mingxiangji

*Xuanyanji* is often regarded as one of the earliest "Buddhist auxiliary texts", and this perception is mainly derived from Liu Yiqing's accounts of his conversion to Buddhism during his later years. However, recent studies have pointed out that *Xuanyanji*, along with Liu Yiqing's other works such as *Shishuo Xinyu* (世說新語) and *Youminglu* (幽明録), were mostly compiled by his subordinates and assistants (Fan 1995; Ning 2000; S. Liu 2008; Zhao 2020). Furthermore, both *Youminglu* and *Xuanyanji* were completed in his later years. Therefore, the extent of Liu Yiqing's personal religious beliefs and the degree to which Buddhism influenced the nature of Xuanyanji remain mysterious.

According to Sano Seiko's research, *Xuanyanji* does not focus on matters related to the underworld like *Youminglu* does. Instead, it includes more content related to Buddhist concepts, such as cause and effect, and karmic retribution. Yet, its approach differs from *Guangshiyin Yingyanji*, which focuses solely on the topic of Guanyin responding to people's prayers, as well as *Mingxiangji*, which includes stories of the underworld as well as the supernatural (Sano 2020, pp. 228–38).

As discussed earlier, the stories found in Xuanyanji and the *MingxiangjI* (See: Robert Ford Campany 2012, pp. 7–17; G. Wang 1999, pp. 2–4) can often emphasize different aspects within the context of their transmission. Among the notable differences, two main variations can be identified:

*Lu*-3: The Official of Wuxing (吳興) Commandery
Direct Source: Possibly Wang Shaozhi (王韶之)
Parallel Texts: *Guanyin Yishu* citing *Guangshiyin Yingyanji*, *Bianzhenglun* citing *Xuanyanji*, *Taiping Guangji* citing *Xuanyanji*, *Fayuan Zhulin* citing *Mingxiangji*, and *Shishi Liutie* citing *Youminglu*.

The general outline of this story is that, during the middle of the Yuanjia era (AD 424–AD 453), the magistrate of Wuxing, Wang Shaozhi, witnessed a large fire engulfing the homes of the people. However, he noticed that the grass hut where a local official lived remained unscathed by the fire. The official had no previous association with Buddhism but had frequently heard Wang Shaozhi talk about Guanshiyin Bodhisattva. During the fire, the official sincerely chanted the name Guanshiyin Bodhisattva and was thus saved. Since this story is narrated by Wang Shaozhi and emphasizes his role in urging the official to have faith, the origin of this story can likely be traced back to Wang Shaozhi.

Various parallel versions of this story highlight the miraculous preservation of the house (see Table 2).

**Table 2.** The differing versions of the Wuxing Official's story.

| Source | Existing Literature | Time | Place | Nonburning Place | Belief |
|---|---|---|---|---|---|
| | Lu | Yuanjia Period (AD 424–AD 453) | Wuxing City | county official's home | recite Guanyin's name |
| *Lu*-3 | *Guanyin Yishu* | | Wuxing City | county official's home | |
| *Xuanyanji* | *Bianzhenglun* | Yuanjia Period (AD 424–AD 453) | Wuxing City | Scripture Hall and Monastic Quarters | scriptures |
| *Mingxiangji* | *Fayuan Zhulin* | 8th Yuanjia (AD 431) | Hetong Puban City | temple, scriptures, and statue | temple, scriptures, and statue |
| *Xuanyanji* (mistaken title) | *Taiping Guangji* | 8th Yuanjia (AD 431) | Hetong Puban City | temple, scriptures, and statue | temple, scriptures, and statue |
| *Youminglu* (mistaken title) | *Shishi Liutie* | 8th Yuanjia (AD 431) | Hetong Puban City | temple, scriptures, and statue | temple, scriptures, and statue |

In terms of time, location, and details of belief, the references to *Taiping Guangji* and *Shishi Liutie* may be erroneous, as both of them overlap too closely with the description in *Mingxiangji*, while *Youminglu* does not record any stories of Buddha or Bodhisattva responded to the prayers. In terms of writing style, the version in *Guangshiyin Yingyanji* is undoubtedly the most realistic and detailed. However, Xuanyanji predates the *Lu* edition, and both mention the same period, suggesting that they may be different versions of the same story that share the same source.

These textual systems also differ significantly in terms of belief: *Xuanyanji* describes the worship of sacred objects and tells stories about visiting the underworld, the *Lu* edition describes the Bodhisattva faith in Buddhism, while the objects of fulfillment in *Mingxiangji* encompass all Buddhist elements. Overall, the *Lu* edition serves as a compromise between the other two. Although the *Lu* edition is the youngest of the three, and its composition references both *Xuanyanji* and *Mingxiangji*, it is challenging to determine whether or not it directly draws from the others due to its intricate storytelling. It is worth noting that the *Lu* edition records the version from the Wuxing region, which aligns with the point on regional memory discussed earlier.

*Lu*-24: Guo Xuan
Direct source: Possibly Guo Xuan's testament

Parallel texts: *Bianzheng Lun* quoting Xuanyanji, *Fayuan Zhulin* quoting *Mingxiangji*, *Shishi Liutie*, *Taiping Guangji* quoting *Bianzheng Lun*, *Shishi Tongjian*（釋氏通鑒） quoting *Seng Shi* (僧史).

The essence of this story is that, in the 11th year of the Yixi era (AD 415), Guo Xuan and Wen Chumao were imprisoned due to being implicated in the indiscriminate killings committed by the Liangzhou magistrate. In prison, they both made vows that, if they were released, they would engage in meritorious deeds in a temple, and later they were saved. The *Lu* edition specifically mentions at the end that Guo Xuan left behind a "testament", which is likely one of the sources for this story.

Among the various versions of the story (see Table 3), the account in *Mingxiangji* is particularly unique. Not only does it not mention Wen Chumao, as seen in other texts, but it is also the only narrative where the vow is made after witnessing auspicious signs. Additionally, the details about the signs and their fulfillment are entirely different. It can be concluded that *Mingxiangji* represents an independent textual system with a unique format.

**Table 3.** The differing versions of the Guo Xuan's story.

| Source | Existing Literature | Governor of Liangzhou | Does Wen Chumao Appear? | He Violates His Oath and Is Struck by Arrow | Ceremony Location | Religious Practices | Supernatural Phenomena |
|---|---|---|---|---|---|---|---|
| | *Lu*-24 | Yang Zijing | yes | no | Upper Mingxi Temple | reciting Avalokitesvara | Dream illumination of Guanyin |
| | *Lu*-24 | | | | | | eight "chi" monk |
| Xuanyanji | Bianzheng Lun | Yang Shoujing | yes | yes | Upper Mingxi Temple | reciting Avalokitesvara | Dream of Bodhisattva |
| Xuanyanji | *Taiping Guangji* | Yang Shoujing | yes | yes | Upper Mingxi Temple | reciting Avalokitesvara | Dream of Bodhisattva |
| *Mingxiangji* | *Fayuan Zhulin* | Yang Siping | no | no | (build a temple) * | reciting Avalokitesvara | Guanyin emitting light |
| | *Shishi Liutie* | Yang Mujing | yes (Wen ChuFa) | yes | West Mingxi Temple | reciting Avalokitesvara | Dream of Bodhisattva |
| Seng Shi | Shishi tongjian | Yang Mujing | Yes (Wen ChuFa) | yes | West Mingxi Temple | reciting Avalokitesvara | Dream of Bodhisattva |

* Taking an oath only after witnessing supernatural phenomena.

Although the story is primarily associated with the Xuanyanji system, there are still differences in details among different versions. For example, although Wen Chumao appears as a co-vower in the *Lu* edition, the breaking of vows portion is omitted. This omission is consistent with the Xuanyanji system and one other.[46] While all versions feature the devotion to Guanyin as a form of belief, the auspicious signs obtained vary. On the other hand, the original *Yingyan* system suggests that the Bodhisattva seen in the dream is not Guanyin, while the *Lu* edition contains a version that involves a belief in the divine monk Ba Chi Dao Ren.[47]

Interestingly, this story is either forged or partially fictional. According to the *Book of Jin*: *Annals of Emperor An*, Liang Zhi Jing, the governor of Liangzhou, was executed as early as July in the second year of the Yi Xi era (AD 406), ten years before the story took

place. Moreover, both individuals who were officials in Liangzhou were imprisoned in Jingzhou instead of being imprisoned locally or at the capital. This suggests that this story was circulating in the context of Jingzhou. This is further supported by their performance of religious ceremonies at Shangmingxi Temple in Jingzhou after their release.

Through recipients of these two stories, it is evident that *Xuanyanji* and *Mingxiangji* not only differ in their dissemination systems but also in the details of the stories and their implied meanings. The *Yingyanji*, which emphasizes specific objects of belief and regional memory, often inherits from the former. This is different from *Xuanyanji*, which mainly recorded *Zhiguai* occurrences centered around the Jingzhou region. *Mingxiangji* not only confirms its devout belief in its preface, but also aims to propagate Buddhism externally. Therefore, its miraculous objects are not limited to specific bodhisattvas or sacred objects, but encompass various Buddhist elements such as pagodas, statues, and scriptures, not to mention that the regions and themes it includes are broader. To a large extent, this breaks the pattern of narrative-style verification and develops into an evangelistic-style record of verification, eventually inherited by the likes of *Mingbaoji,* etc.

*3.3. Categories of Zhiguai, Yingyan, and Gantong Dissemination Systems*

Komina Ichirō once classified Six Dynasties literature into two categories: *Zhiguai* and proselytization (Komina 1982, pp. 415–500). However, this classification did not include forms of dissemination found in Buddhist texts such as biographies of monks, making it incomplete. Furthermore, he did not discuss differences in the dissemination of stories between different types of texts. Therefore, the following section will re-examine the relationship between the classification of texts and the transmission of stories in different systems.

Previous research has focused on the evolutionary relationship between *Zhiguai* (annals of the bizarre) and *chuanqi* (miracle) literature. However, few scholars have noted that these supernatural stories also use different systems of writing depending on the authors' perspective. Due to these different perspectives, we can associate *Zhigui* writings with non-Buddhist believers, *Yingyan* writings with lay practitioners who have Buddhist beliefs, and *Gantong* writings with Buddhist monks (see Table 4).

**Table 4.** The different systems of miracle stories.

| Non Buddhists | Laymen | Laymen | Monk | Monk, Laymen |
|---|---|---|---|---|
| *Zhiguai* | Descriptive style *Yingyan* | Propagative style *Yingyan* | *Gantong* | Compilation |
| *Xu Gaoseng Zhuan* *Shuyiji* *Zhenyiji* | *Xuanyanji* *Fu* edition *Zheng* edition | *Mingxiangji* *Lu* edition | *Fahua Yishu* *Guanyin Yishu* *Fahua jing Wenju* *Fuzheng Ji* (Biographies of monk or nuns) | *Fayuan Zhulin* *Taiping Guangji* |

If the purpose of the *Zhiguai* system is to preserve extraordinary and unusual events, then the purpose of the *Yingyan* system is to facilitate the exchange of faith-based stories among fellow Buddhist believers. Based on their objectives and source of information, *Yingyan* writings can be further divided into two types: the descriptive style of *Yingyan* that emphasizes regional memories and bears resemblance to *Zhiguai*-style writings, and the propagative style *Yingyan* writings which have a broader range of source materials and stronger emphasis on spreading Buddhism. Additionally, there are also *Gantong* writings, which include biographies of monks, exegesis of scriptures, and *Yingyan* narratives composed by monks.[48]

The elements selected and discarded among different systems are illustrated through the stories of Zhu Changshu (*Guang*-1) and Gao Xun (*Xi*-15):

*Fu*-1: Zhu Changshu

Parallel Texts: *Guanyin Yishu*, *Fahua Yishu* citing *Fu* edition, *Bianzheng Lun* citing *Jinlu* (晉録), *Mingxiangji*, *Fayuan Zhulin* citing *Mingxiangji*, *Fahua jing Wenju Fuzheng Ji*, *Shishi Liutie* citing *Jinlu*, *Mingxiangji*, *Fahua Zhuanji* citing *Fayuan Zhulin*, *Taiping Guangji* citing *Bianzheng Lun*.

     Transmission Systems:

1.  *Fu* edition --> *Jinlu, Mingxiangji* --> *Bianzheng Lun, Shishi Liutie* --> *Taiping Guangji*.
2.  *Fu* edition --> *Mingxiangji* --> Treasury of the Dharma, *Fahua Zhuanji*.
3.  *Fu* edition --> *Guanyin Yishu, Fahua Yishu*, Lotus Sutra *Fuzhengji*.

     The *Zhiguai* system originated from *Jinlu* and *Mingxiangji*. Compared to the *Yingyan* system, the content in *Zhiguai* represents a localized version of the Wu region, likely based on the narrative in *Jinlu*.[49] This system was first recorded in *Bianzheng Lun* and was later included in *Shishi Liutie* and *Taiping Guangji*. The characteristic of this system is that the texts are mostly compiled works, often directly copied with little modification (see Table 5).

**Table 5.** The differing versions of the Zhu Changshu's story.

| | Source | Religious Practices | Come From | Place | Chanting the Guanyin Sutra with Reverence | Order Family Do Not Bring Anything |
|---|---|---|---|---|---|---|
| | *Fu* edition | chanting scripture | Xiyu | Luoyang | yes | yes |
| *Zhiguai* | *Bianzheng Lun* | recite | India | Wuzhong City | yes | no |
| | *Shishi Liutie* | recite | India | Wuzhong City | yes | no |
| | Taiping Guangji | recite | India | Wuzhong City | yes | no |
| *Yingyan* | *Fayuan Zhulin* | chanting scripture | ancestor come from Xiyu | Luoyang | yes | yes |
| | *Fahua Zhuanji* | chanting scripture | ancestor come from Xiyu | Luoyang | yes | yes |
| *Gantong* | *Guanyin Yishu* | recite | | Luoyang | no | no |
| | *Fahua Yishu* | recite | Xiyu | | no | no |
| | *Fahua jing Wenju Fuzheng Ji* | recite | | Luoyang | no | no |

     The *Yingyan* system is based on the *Fu* edition and is included in *Mingxiangji*, as well as *Fayuan Zhulin* and *Fahua Zhuanji*. This system has the largest overlap with the *Fu* edition. Both *Mingxiangji* and *Fayuan Zhulin* are texts with strong missionary undertones, but due to the way they were compiled, they have not undergone significant modification.

     These systems were mostly authored by laypeople or heavily influenced by lay authors and emphasize religious practices such as reciting scriptures. They focus on the manifestation of religious behaviors and demonstrate the repetitive nature of these actions. The *Gantong* system, on the other hand, within the context of religion, emphasizes the mystery of things themselves and focuses on explaining the immediacy of fulfillment and the sanctity of the object of belief. Therefore, it tends to modify actions to better reflect the names of the objects of belief. At the same time, the *Gantong* system tends to simplify and dehistoricize narratives, either by omitting time references or changing the historical timeline to match that of the event, making it the most extensively modified textual system (see Table 6).

*Lu*-15: Gao Xun

Direct sources: *Shuzhengji* and *Zhenyiji*

Parallel texts: *Guanyin Yishu* citing *Fu* edition, *Bianzheng Lun* citing *Xuanyanji* and *Xusou Shenji*, *Fahua jing Wenju Fuzheng Ji*, *Taiping Guangji* citing *Xuanyanji*, *Sanbao Ganying Yao Luelu* citing *Xuanyanji*.

**Table 6.** The timeline of varying versions of Guo Xuan's story.

|  | Source | Time of Entering China | Time that the Story Happened |
|---|---|---|---|
|  | *Fu* edition | Yuankang reign period (291–99) |  |
| *Zhiguai* | *Bianzheng Lun* |  | Jin dynasty |
|  | *Shishi Liutie* |  | Jin dynasty |
|  | *Taiping Guangji* |  | Jin dynasty |
| *Yingyan* | *Fayuan Zhulin* | Yuankang reign period (291–99) |  |
|  | *Fahua Zhuanji* | Yuankang reign period (291–99) |  |
| *Gantong* | *Guanyin Yishu* |  | Yuankang reign period (291–99) |
|  | *Fahua Yishu* |  |  |
|  | *Fahua jing Wenju Fuzheng Ji* |  | Yuankang reign period (291–99) |

This particular system has already been analyzed above. The two systems differ not only in names used for the main characters, but also in their recorded year of death. The *Xuanyanji* system states the fifth year of the Taiyuan Era (AD 370), while the *Guangshiyin Yingyanji* and subsequent *Gantong* systems state the seventh year of the Taiyuann Era (AD 372). The transmission context is as follows(Figure 2):

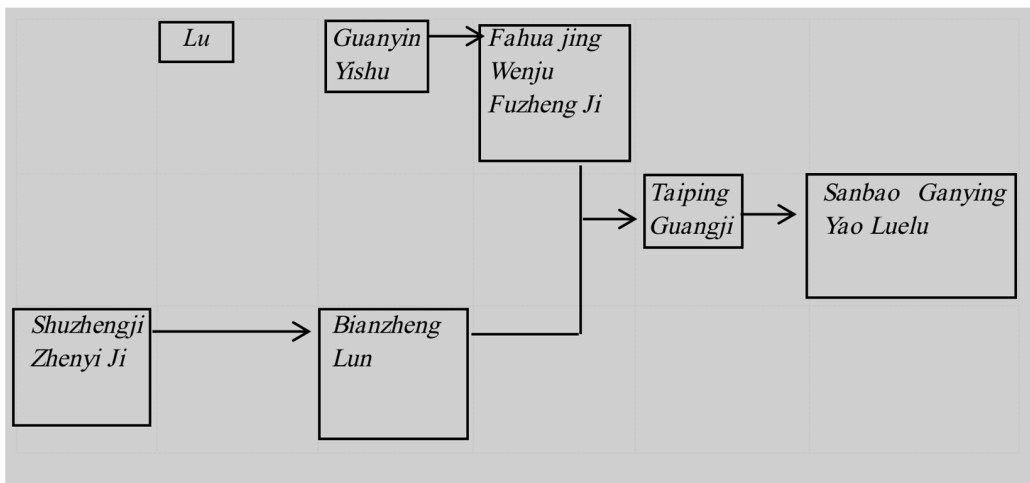

**Figure 2.** The transmission of the Gao Xun's story.

The *Zhiguai* system presents "Buddhist deities" (佛神), which are more realistic and imbued with local elements, while also emphasizing their divine power. In contrast, the *Yingyan* system in the *Lu* edition highlights the compassionate power of Guanyin Bodhisattva in the Western Pure Land, who can save people. The *Gantong* system, on the other hand, presents a brief narrative, emphasizing beheading and the shattering of a sword, as well as the act of selling oneself and one's wife to support the monks.

The practice of removing or altering specific timeframes is also commonly seen in *Gantong* writings, particularly in stories involving miraculous monks such as Shi Daojun (*Xuan*-6),[50] Shi Senghong (*Xuan*-22),[51] and Guo Xuan (*Xuan*-24). Among various records, only the *Shishi Yaolan* (釋氏要覽) does not include any temporal references.[52] However, this practice is mainly observed in the narration of miraculous monks. In the "Yi Jie" chapter in *Zhu Fa Yi*, though there are fantastical accounts of miracles, his biographical content,

copied mainly from *Shamen Tanzongsiji*, includes records of studying under the famous monk Shengong, delivering sermons, and having the emperor build a temple and burial site for him. Therefore, this biographical style prioritizes realism and the preservation of time and place, while the accounts of miracles become secondary and supporting narratives.

Differences in miracle stories among different systems are not uncommon. When Sun Shangyong studied the legend of "Fish Mountain and the Brahman Chant" (魚山梵唄), he pointed out that this story was first recorded in *Yiyuan* (異苑), which included two different versions: one about the Brahman chant and the other about Taoist illusionary footsteps. However, only the version of the Brahman chant survived and developed into various interpretations, such as transferring merits through chanting scripture, or performing the Brahman chant (S. Sun 2008, pp. 144–48).

Due to the long history of dissemination, many accounts of miraculous stories from different systems have been lost. The existing versions are primarily preserved in texts with a Buddhist perspective, giving one the impression of coherence and consistent views. Working to clarify the different dissemination contexts of miracle stories helps one grasp the nature of different texts and their underlying historical background. This, in turn, helps one break away from the traditional narrative of Buddhist history, allowing the stories to be examined within their original temporal and spatial contexts. Stories are disseminated by multiple actors in the process of being transmitted, and the stories themselves are constructed by participants such as eyewitnesses, storytellers, transmitters, recorders, as well as local memories. Hence, it becomes apparent that multiple actors with diverse perspectives are involved in disseminating, rewriting, and piecing together these stories. By analyzing and dissecting these various elements, these stories can be better placed within their original historical contexts and more accurately understood.

## 4. Conclusions

From this analysis, it may be observed that the original intention of *Yingyanji* was not to propagate Buddhism, but rather to serve as a collection of religious stories circulated among the scholarly elite. It bears the dual characteristic of familial and regional memory. Different *Yingyan* writings demonstrate variation in their perspective, have different purposes, and use different sources. They may not necessarily have been written with a clear intention for use in propagation. Thus, the generalization of *Shishi Fojiao zhi shu* (Books for Assisting Buddhist Teaching) ought to be reconsidered. This concept should be examined under the spectrum of *Zhiguai* and *Gantong*, while the influential relationship between stories intended "to preach to people and lead them to conversion" and *Yingyan* writings ought to be inverted.[53]

Different texts demonstrate variation in their attitude toward and handling of supernatural stories. To substantiate their claims, both *Zhiguai* and *Yingyan* writings often adopt a biographical style of narration while including descriptions of the scenes. *Guangshiyin Yingyanji* contains many stories related to the author's personal acquaintances, including many details and sources unrelated to the main story. The *Gantong* system, however, abandons this realistic narrative technique. The narratives of the monk biographies rely primarily on documentary sources such as scripture or temple records. Supernatural narratives are imposed as separate story blocks, with less focus on time, place, and source.

Due to the limited availability of visual images and documentary materials from the Six Dynasties period, our understanding of Buddhist beliefs during that time has often been limited to a few objects of worship, such as Guanyin and Amitabha. However, through different versions of miracle stories and the stratification of their dissemination, we can see a more diverse religious landscape. Just because certain objects of worship or scriptures are less frequently mentioned or recorded does not mean their value is lesser. The fact that a story or piece of scripture has been altered or deleted only goes to demonstrate its significance during that period. Only by continuously excavating obscured narratives can we better reconstruct religious landscapes in their historical contexts.

**Funding:** This research received no external funding.

**Institutional Review Board Statement:** Not applicable.

**Informed Consent Statement:** Not applicable.

**Data Availability Statement:** Not applicable.

**Conflicts of Interest:** The author declares that there are no conflict of interest.

## Abbreviations

T Taishō shinshū daizōkyō 大正新脩大藏經 (Taishō edition of the Buddhist canon). Ed. Takakusu Junjirō 高楠順次郎 et al. 100 vols. Tokyo: Taishō Issaikyō Kankōkai, 1924–1935. X Manji Shinsan Dainihon Zokuzōkyō新纂卍續藏 (New Compilation of Buddhist canon) Ed. Kawamura Kōshō 河村孝照 et al. 90 vols. Kyoto: Zōkyō shoin, 1975–1989. B Supplement to the Dazangjing大藏經補編 et al. 36 vols. Taipei: Lan Jifu 1985.

## Notes

[1] The term "Buddhist auxiliary texts" was proposed by Lu Xun (1881–1936) in the early 20th century. Lu Xun is considered the founder of modern research on ancient Chinese novels and one of the most important researchers in ancient Chinese literature. "Buddhist auxiliary texts" was the earlier definition for Buddhist miracle stories. It emphasizes the difference from normal Chinese novels and the purpose of Buddhist proselytizing. This concept was widely accepted by later researchers.

More broadly, Lu Xun defined "Buddhist auxiliary texts" as a kind of *Zhiguai*. Reconsidering the definition of "Buddhist auxiliary texts" in this approach helps us understand the complexity of *Zhiguai* and the development of medieval novels.

[2] The main research can refer to the work of Liu Huiqing, Leng Yan, and others. In addition, there are some influences on the thought of reincarnation and other genres of Buddhism, which have also been discussed by scholars. (X. Li 2015; H. Liu 2013, 2019; Jin 2016, pp. 118–21; Leng 2019; Huang 2013, pp. 119–20; Peng and Zhou 2019, pp. 145–51).

[3] *Changdao* refers to a preaching procedure in Buddhist rituals, where the speaker uses plain language and Vipāka, or fateful stories, to explain the principles and teachings found within Buddhist scriptures to the audience.

[4] Research on the relationship between "preach[ing] to people [to] lead them to conversion" and "Buddhist auxiliary texts" primarily relies on evidence from later Tang Dynasty commentaries, variant texts, and sermon texts to make inferences. This may be related to the perspective of considering the Jin and Tang Dynasties as a unified entity in relevant studies. Among them, the discussion by Li Xiaorong is the most comprehensive. He believes that oral chanting, accounts of miraculous experiences by laypeople, and extensive records of knowledge are the three creative sources of these stories, which broadly summarize the various motives for the creation of the "Books of Buddhist Auxiliary Teaching". However, he still does not explain why accounts of miraculous experiences by laypeople predate those by monks. Related studies can be referred to: (Hu and Zhou 2013, pp. 64–70; G. Zhang 1995, p. 10; E. Zhang 2007, pp. 43–46; X. Li 2015, pp. 48–57).

[5] Traditional studies on *Zhiguai* generally adopt a genre-based approach and identify a category of works within the supernatural "tale" genre that serves the purpose of propagating Buddhism. These works are often referred to as "Books of Buddhist Auxiliary Teaching" or "Buddhist Propagation Fiction". They are considered an exception or supplement within the classification of supernatural tales and are sometimes even regarded as the lowest form of supernatural tales. From Lu Xun's classification of *zhi ren* (lit. "people of ambition") and *zhiguai* (lit. "strange and extraordinary"), to Liu Yeqiu's later establishment of the tripartite division of "geographical knowledge and natural history", "narratives of gods and spirits", and "miscellaneous histories and records", the "Books of Buddhist Auxiliary Teaching" have not obtained an independent status but rather serve as an extension or supplement to the "strange tale" genre. ( Y. Liu 1987, p. 83).

[6] The study of pre-Tang supernatural tales has been extensively conducted in academic circles. Early scholars such as Lu Xun and Yu Jiaxi laid the foundation for research methods centered around the typology of stories, characters, and historical sources. In Wang Guoliang's article "A Hundred Years of Research on *Zhiguai*: Tracing the Shift in Literary, Historical, and Cultural Studies", he points out that early research primarily focused on the compilation of catalogs, versions, and authentication of documents. During this period, the emphasis was on the compilation of supernatural stories and the definition of the genre. However, in the 1980s and 1990s, the judgments regarding the relationship between legend and *Zhiguai*, and whether *Zhiguai* should be considered literary creations, began to be questioned. Scholars started to reflect on the ambiguity of the definition of *Zhiguai*, and the research direction shifted toward areas such as genre studies and cultural history. Emphasis was placed on literary imagery, linguistic analysis of terms and expressions, the origins and development of fictional narratives, and the process of story transmission and the social environment in which they were received.   The first two aspects are not directly related to this discussion and will not be elaborated upon here. Regarding the issue of origins, Zhang Qingwen believes that *Zhiguai* is a transformative inheritance of disaster stories that served as political warnings, stripping away their original political and ethical functions and focusing on highlighting the supernatural and entertainment aspects of the stories. Wang Yao, on the

other hand, argues that the origins of *Zhiguai* lie in the "exaggerated language" of *Fangshi* (magicians and alchemists). Scholars such as Li Jianguo and Wang Xin emphasize that *Zhiguai* is a product of the intellectual trend of natural history during the Wei and Jin periods, which incorporates the techniques and experiential knowledge of *Fangshu* (divination and magic). They emphasize its technical and knowledge functions. Scholars like Yao Xiaoyou and He Jin focus on discussing the origins of *Zhiguai* in unearthed documents. Regarding the discussion of story transmission and cultural history, the most representative research is by Komina Ichirō. He discusses stories within the context of storytellers, providing detailed discussions on the transmission of different *Zhiguai* texts. Wang Xin, on the other hand, pays attention to the intertextuality between geographical works and *Zhiguai*, pointing out that the caves described in Six Dynasties regional records are not objective records of natural geography, but rather supernatural sacred sites and memorials packaged in various miraculous events. Wei Bin discusses the existing supernatural narratives of An Shigao, which mix various descriptions from different sources and demonstrate the complexity of the narrative purposes of supernatural stories. Looking at previous research, recent studies continue the genre-based discussions, analyzing the origins of the novels and focusing on the nature of *Zhiguai* works. On the other hand, there is a growing interest in exploring the contextual aspects of story generation and transmission, paying more attention to individual themes or stories that span across different texts. However, if the latter approach is divorced from genre discussions, it neglects the intention behind the compilation of *Zhiguai* texts. If the former approach does not consider the themes and narrative qualities, it may fall into the misconception of viewing *Zhiguai* works as closed entities. Therefore, there is a need for organic integration between the two approaches. (Xie 2011; Ning 2017, pp. 37–41; Q. Zhang 2000, pp. 11–13; Jiang 1996; Y. Wang 1998, p. 134; X. Wang 2018b, pp. 128–40; 2017, pp. 137–45; Wei 2012, pp. 39–48)

7  Regarding the research overview of three *Guangshiyin Yingyanji*, you can refer to the compilations by Dong Zhiqiao and Yu Junfang. Please see: (Dong Zhiqiao, 2002, pp. 4–9; Yu 2000, pp. 152–53). Other relevant sources include: (Sano 2020, pp. 238–67; Yu 2000, pp. 167–82; Gu 2015; Wu 2007, pp. 123–27).

8  T 2122, 53. p. 418c6-7.

9  The internal transmission mentioned here is different from Taoism, which emphasizes secretive transmission and emphasizes the passing down of teachings from master to disciple, often accompanied by rituals. This can be compared to the sharing of personal religious experiences in Christianity, where worshippers share their personal experiences during worship gatherings or collective prayers. The main purpose is to validate and share their revelations, thus strengthening their religious beliefs. As for examples from the literature, one can refer to the *Guixin pian* chapter in *Yanshi jiaxun*顏氏家訓, where Yan Tui recorded several stories related to killing and its consequences. The purpose here is undoubtedly different from that of the *Yuanhun Zhi*, also authored by Yan Tui. The former serves as a warning to the family about karmic retribution, consolidating and inheriting the family's beliefs, while the latter has a slight missionary implication. This serves a function similar to the one described by Huiyuan when discussing the spread of stories about miraculous manifestations of the Buddha, where he said, "Every thought of strange phenomena is to strengthen their sincerity" and "to verify the myriad paths of gods". Furthermore, Yu Junfang repeatedly emphasizes the similarity between *Yingyanji* and stories of filial piety from the Han dynasty, even suggesting that some stories may have been adapted from filial piety stories. Therefore, *Yingyanji* also has certain local elements. Nankai's research on the transmission of filial piety stories also illustrates the characteristics of the internal transmission of such miraculous stories. Please refer to: (Yu 2000, pp. 167–79; Knapp 2005).

10  The three editions of *Yingyanji* are referred to as the *Guanshiyin Yingyanji* (*Fu* edition), Xu Guangshiyin *Yingyanji* (*Zhang* edition), and *Xi Guanshiyin Yingyanji* (*Lu* edition) by the author. The number following the edition refers to the number rank of the edition. For instance, *Fu*-5 refers to the fifth story in the *Xi Guanshiyin Yingyanji (Fu* edition*)*.

11  This biography is also mentioned in Wang Manying's (王曼穎) reply to Huihuan (慧皎). The further research of Zheng family, see (S. Li 2018).

12  According to Li Jianguo, the earliest Buddhist monk-authored "Buddhist auxiliary texts" is Tan Yong's *Soushen Lun*, written during the Northern Wei dynasty. However, the content of this book was not detailed. It was not until the Sui dynasty that similar works began to appear, which was much later than the flourishing layman works of the Qi and Liang dynasties.

13  Both positions were held during the end of the Jin dynasty. Since Zhang Yan was active during the Yuanjia元嘉 era, he likely grew up in Jingzhou during his childhood.

14  The explanation for this citation style is mentioned in Note 16. (*Zhang*-7) refers to the seventh story in Zhang Yan's *Xu Guangshiyin Yingyanji*.

15  Zhang Yan mentioned the latest reign title as Yi Xi義熙 and the latest event as the rebellion of Sun En孫恩之亂.

16  *Lu*-3 is another example of regional memory transfer. It was originally recorded in a story that took place in Wuxing during the Yuanjia period, but in *Xuanyanji* it was set in Hedong (河東), Puban (蒲阪), during the eighth year of the Yuanjia era (AD 430). A later and more representative example of regional memory construction is a series of legends related to Liu Sahhe in Bingzhou. For example, his propagation of Buddhism in Hexi (河西) led to the attribution of his origin to Danyang (丹陽) in Danzhou (定陽) after the Western Wei period. In the early Tang period, legends emerged in Dunhuang about Liu Sahhe (劉薩訶) bestowing scriptures in the Mogao Caves (莫高窟). For more information, refer to (Shang 2007, pp. 65–74). I intend to discuss this phenomenon further in another article.

[17] For example, Xu Yi (*Zhang*-1) shared his personal experience with Huiyan, who then relayed the story to Zhang Yan. The story of a certain individual during the Yi Xi era (*Zhang*-9) was told to Faxiong by Mao Dezhu (毛德祖) and later conveyed to Zhang Yan. The incident in North Pengcheng (*Zhang*-13) was directly heard by De Cangni and later conveyed to Zhang Yan by Shi Huiqi, a disciple of Lu Gao. This may be due to the convenience of travel for monks during that time, allowing for an easier exchange of stories between the North and South.

[18] T 2110, 52. p. 539b.

[19] Although the practice of offering lamps to the Buddha (施燈供佛) is inherent to Buddhism, the combination of seven lamps may have been influenced by local beliefs. The earliest mention of the seven lamps in Buddhist scriptures can be found in the mid-Southern Liu-Song dynasty apocryphal sutra *Foshuo Guanding Jing* (佛說灌頂經), specifically in the seventh and twelfth volumes. In the seventh volume, the "seven lamps" correspond to the summoning of seven divine kings and are unrelated to the practice discussed here. However, in the twelfth volume, "seven lamps" are part of the method of the Banner of Prolonging Life, emphasizing the use of a five-colored divine banner and seven-tiered lamps (with one lamp per tier). Scholar Wu Xiaoshao has discussed the Banner of Prolonging Life and pointed out its influence on local beliefs related to karmic retribution and blessings. He also mentions a record in the *Chu Sanzang Ji Ji* (出三藏記集) titled "Record of the Seven-Tiered Lamp Dispelling Suffering", which references the *Foshuo Guanding Jing*, indicating that the practice described in *Xuanyanji* is derived from the practice in the *Foshuo Guanding Jing* and later became a specific practice within the Medicine Buddha cult. On the other hand, in Taoism, although the practice of lighting lamps is borrowed from Buddhism, the concept of seven lamps predates it. During the Jin and Song dynasties, the *Shangyuan Jinlu Jianwen* (上元金籙簡文) had already mentioned the practice of lighting seven lamps, which corresponds to the seven souls or the seven sets of parents in human life. The function of the seven souls was to "pacify the spirit and eliminate calamities, control the souls and remove evil", which is remarkably similar to the concepts in the "Sutra of Empowerment and Elimination of Transgressions for Attaining Salvation from Life and Death". Therefore, the practice of lighting seven lamps may have originated from Taoism. Refer to: (Wu 2010, pp. 128–31, 201–17; Lü 2007, pp. 10–12). Special thanks to fellow student Yuhang Chen for pointing out that the *Shangyuan Jinlu Jianwen* is the earliest reference to the practice of lighting lamps in Taoism.

[20] As Natalie Zemon Davis discussed in her analysis of pardon letters, these supposedly factual accounts often contain elements of *Zhiguai*. Despite being expected to provide an accurate description of the case, various parties including the petitioner, their representatives, royal notaries, and secretaries were involved in the creation and embellishment of these narratives. See: (Davis 1987, pp. 4–5). On the other hand, some scholars have begun to reflect on the position of "fictional writing". For instance, when analyzing geographical records and tales of the strange, Wang Xin points out the intertextuality of these two types of texts and considers them as a form of describing the extraordinary rather than being entirely original. See: (X. Wang 2017, pp. 137–45; 2018a, pp. 119–28).

[21] For example, in *Xì*-46, Shi Kaida encounters a tiger; in *Xì*-47, Pei Anqi encounters a wolf and is rescued; in *Xì*-59, Shi Senglang encounters a tiger, and there is also a record in *Songshu* where Wang Yinan escapes. All of these stories share a similar narrative structure, with the protagonists encountering and being saved or guided by fierce animals while fleeing to the South. This repetition of similar narratives suggests their recurrent usage.

[22] According to the records in the *Songshu*, there was indeed a magistrate of Wuxing named Zhang Chong. The same story is also mentioned in *Fayuan Zhulin*. See: (Shen Yue, Songshu, 1974, pp. 2247–48; T 2122, 53. p. 659).

[23] Both *Fayuan Zhulin* and *Ji Shenzhou Sanbao Gantonglu* include the story of Zhu Lingshi being rescued in Liaodong, with the appearance of a divine cup at the end. The historical records do not mention the incident of Zhu Lingshi in Liaodong. *Fayuan Zhulin* attributes this story to *Gaoseng Zhuan*, but neither of the biographies of the monks in question mentions it. This story is likely a combination of legends involving sitting on a cup and crossing water.

[24] T2063, p. 939.

[25] According to the details mentioned in the letter from Wang Manying to Huijian and the preface of *Gaoseng Zhuan*, Cao Daoheng even suggests that Wang Manying is the son of Wang Yan. If this theory is correct, then *Mingxiangji* and *Buxu Mingxiangji* can be seen to some extent as collections of internally transmitted stories. See: (Cao 1992, p. 27).

[26] The *Fu* edition also only provides two indications of the time of occurrence, but the sources mentioned or other time points indirectly determine the timeframe of the stories.

[27] One remaining story is very short, while the other two are related to the renowned general Mao Dezhu in the period between the Jin and Song dynasties.

[28] For example, the story of "Pengcheng Widows" in the *Lu* edition includes three different versions, the story of "Gao Xun" cites three different sources, and *Guangshiyin Yingyanji* draws inspiration from both Xuanyanji and Mingxiangji.

[29] Regarding the compilation policy of *Mingxiangji*, please refer to Sano Seiko's work: "Recording the Strange: The Birth and Development of Six Dynasties Zhiguai", pp. 233–38. As for the classification of the contents of Mingxiangji, Wang Guoliang categorized the stories into eleven thematic categories, while Hou Xudong divided them into seven categories, encompassing various aspects of Buddhist beliefs. See: (G. Wang 1999, pp. 27–45; Hou 2018, pp. 44–45).

30    The term *Yingyan* originates from the *Guangshiyin Yingyanji*. Although the concept of *Yingyan* is not exclusive to lay practitioners, the concept of *Yingyan* refers to the purposefulness of their prayers and the immediacy of the response from the object of their faith. Therefore, choosing *Yingyan* represents the perspective of lay practitioners.

31    The term *Gantong*, derived from the *Gantong* chapter in *Xu Gaoseng Zhuan*, is intended to distinguish itself from the system of supernatural and strange phenomena outside of Buddhist teachings. The reason why I prefer *Gantong* over *Ganying* (感應) is that *Ganying* is a term used in Buddhist and non-Buddhist contexts, representing miraculous replies from any deity or god. On the other hand, *Gantong* is usually used to describe miraculous replies related to filial piety and the Buddha. Therefore, *Gantong* is a more suitable term to represent the stories written by monks, reaffirming the close relationship between filial piety stories and Buddhist stories.

32    The analysis of another story can be found in Sections 2 and 3.

33    Although Lu Xun, Li Jianguo, and Hirata Masashi did not include this entry in their compilation of *Shuyiji*, they did not explain it and simply attributed it to *Mingxiangji*. However, based on the analysis in this article, the narrative of *Zhiguai* represented by *Shuyiji* differs from the texts recorded in other systems. Therefore, it is likely to be reliable.

34    T 2122, 53. p. 409.

35    According to Makita Tairyō, he suggests that the references in *Fayuan Zhulin* are an abridged version of *Mingxiangji,* but does not provide evidence to support this claim. See: (Makita Tairyō, 1970, p. 82).

36    In the first half of the entry on *Mingseng Zhuan,* Chao quotes from Shamen Tanzongsiji, describing Zhufa Yi's early years of studying under Shengong, his preaching, and socializing in the capital, as well as the establishment of Xinting Temple by Emperor Xiaowu of Jin in his honor after his death. The second half of the entry on Zhufa Yi overlaps with the content of *Guangshiyin Yingyanji*.

37    T 2122, 53. p. 988.

38    *Taiping Guangji* also supports the claim of the seventh year of Taiyuan (AD 372). However, *Mingseng Zhuan* and *Gaoseng Zhuan* adopt the account of the fifth year of Taiyuan (AD 370), as mentioned in *Shamen Tanzongsiji*.

39    According to the record in the "Treatise on Literature" chapter in *Sui Shu* (隋書經籍志), Dai Zuo wrote the works *Zhen Yi Zhuan* and *Xi Zheng Ji*. This incident occurred in Rongyang (滎陽) and is unrelated to the Western Expedition; thus, it should be the account mentioned in *Zhen Yi Zhuan*.

40    This refers to the chanting or meditation practice of "Mahayana Pure Land Belief", which is primarily found in contemplative scriptures such as *Banzhou Sanmei Jing* (般舟三昧經) or *Guan Wuliang Shou Jing* (觀無量壽經). For instance, in *Banzhou Sanmei Jing*, it states: "The Buddha said, 'Bodhisattvas in this land should contemplate the name of Amitabha Buddha exclusively, and through exclusive contemplation, they will be able to see him'". The Lotus Sutra also contains similar expressions: "If you wish to attain the five kinds of clairvoyance, you should abide in a quiet place, concentrate your thoughts, and contemplate the Way. By doing so, you will be able to understand it". T417, 13. p. 899a; T 263, 9. p. 86a-b.

41    The *Fayuan Zhulin* records the story of Zhu Changshu possessing a family heirloom relic, which is also mentioned in the *Ji Shenzhou Sanbao Gantonglu*. Therefore, it is likely that this account had already appeared before the Tang Dynasty. In the story, Zhu Changshu's son is also a Buddhist monk who frequently desires to return to secular life but repeatedly gives up due to the miraculous powers of the relic itself. Such stories of monks having children and desiring to return to secular life, which deviate from the ideal, are naturally not accepted or adopted.

42    T 2110, 52. p. 539a.

43    T2122, 53. p. 484b.

44    T2122, 53. p. 678b.

45    Special thanks to Dr. Qiye Xie for his reminder regarding the writing style of *Xuanyanji*.

46    The available sources for monastic history texts are primarily Wang Jin's *Seng Shi* from the Liang dynasty and the *Da Song Seng Shi Lüe* (大宋僧史略) from the Southern Liu-Song dynasty. This system identifies the Upper Mingxi Temple (上明西寺), established during the Eastern Jin Dynasty, as the West Mingxi Temple (西明寺), established during the Tang Dynasty, indicating that it is unlikely to have originated from Wang Jin's *Seng Shi*. Moreover, the *Da Song Seng Shi Lüe* was written later than the two texts from this system.

47    Even if we convert to the shortest measurement of the Southern Dynasty, which is approximately 24.7 cm, this monk would still be nearly two meters tall, much taller than the average height, not to mention that measurement units during the Northern, Sui, and Tang Dynasties were approximately 30 cm. Therefore, the term "eight *chi*" (approximately 2.4 m) in historical records, biographies of monks, and novels, are often accompanied by adjectives praising someone's extraordinary temperament. The legend of the tall monk, "Eight *Chi* Dao Ren", seems to have been popular at that time. Both the *Ji Shenzhou Sanbao Gantong lu* and *Fayuan Zhulin* contain stories about Di Shichang (抵世常) and mention the appearance of a monk with supernatural powers who would manifest in an eight-*chi* form during such occasions. See: Qiu Guangming, "A Study of Chinese Weights and Measures throughout History", Science Press, 1992, p. 520; T2106, 52. p. 432a; T2122, 53. p. 492b.

48    During the compilation of monastic biographies, miscellaneous records, and accounts of supernatural phenomena by Buddhist monks, it can be observed that a significant number of details are being altered. This issue is being addressed in a separate article

by the author. Please refer to: "Historical Sources and Compilation of Medieval Monk Biographies: Focusing on Supernatural Stories" (to be published).

49 There were three kinds of records titled *Jin Lu*, which were written by Yu Yu, He Fasheng, and Zhu Fazu, all of which are now lost. *Bianzheng Lun* quotes *Jin Lu* three times, including instances of Xie Hui breaking a pagoda and receiving retribution, as well as Wang Ning's wife encountering her deceased child and being persuaded to convert. Therefore, it cannot be the work of Yu Yu, which recorded events during the Western Jin Dynasty. Furthermore, Zhu Fazu's records as a Buddhist scripture generally does not contain precedents for recording secular stories of spiritual responses, and these individuals are also unrelated to Buddhist translation. Therefore, this book is likely the *Jin Lu* by He Fasheng, written during the Liu-Song dynasty. In this context, its records should be slightly later than *Guangshiyin Yingyanji*.

50 Regarding the narrative technique of removing a temporal context from *Gaoseng Zhuan*, Shinohara Koichi specifically discusses the case of Shi Daojiong. See: (Shinohara 1988, pp. 119–28).

51 Among various records, only the *Gaoseng Zhuan* simplifies the year Yi Xi 11th (338 CE) to the end of the Jin Dynasty.

52 Among various records, only the *Shishi Yaolan* does not include any temporal references.

53 In previous studies, one was more inclined to consider anecdotal texts as secularized versions of stories originally preaching or leading the chants by monks. For example, Zhang Eping straightforwardly states that the transformation of "books on ghosts, gods, and strange phenomena" and the rise of "books assisting Buddhist teachings" were directly influenced by the *Changdao* practices of monks since the Eastern Jin Dynasty. This assertion is mainly supported by two pieces of evidence. First, it is believed that certain stories can trace their origins to Indian and Central Asian parables, as pointed out in the studies of Lu Xun and Li Jianguo. However, these stories are often not the main contents of these anecdotal texts. More importantly, works such as *Xianyu Jing* (賢愚經; Sutra on the Wise and Foolish) and *Baiyu Jing* (百喻經; Sutra on a Hundred Parables) do not primarily consist of foreign stories, but instead rely on "recent events" that occurred locally. When "good and evil, calamities and blessings, and signs and portents" (古今善惡禍福徵祥) are listed in *Fayuan Zhulin*, there is no mention of *Xianyu Jing* or *Baiyu Jing*. Hence, the content of *Changdao* primarily consists of local stories or personal accounts heard by the monks, rather than preexisting Buddhist parables. Yet, on the other hand, from the records found in current accounts of *Zhiguai* or anecdotal texts, it is mentioned that some stories were acquired during the process of preaching or leading chants. Clearly, this inference lacks direct evidence. Whether in biographies of monks, commentaries on Guanyin, accounts from Qingliang, or scriptures such as the Lotus Sutra, these chant texts predominantly adopt supernatural stories with local settings recorded by laypeople, which can even be traced back to *Zhiguai*. Furthermore, there are examples of instructors initially being laypeople, and the rise of anecdotal text and chant practices occurred roughly simultaneously with the sources of those anecdotal stories being provided by laypeople. All of these factors indicate that the creation of anecdotal stories from a layperson's perspective was likely parallel to the proselyting practices of monks, and the latter may have drawn more from the former. For references to monks' use of "recent events" in their propagation of the Dharma, please refer to: (Hou 2018, p. 48).

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
