# Peer review of "Miracle Stories in Motion—On the Three Editions of Guangshiyin Yingyanji"

_religions, doi:10.3390/rel14091114_

Round 1
Reviewer 1 Report
This essay shows much promise. The author advances an interesting theory that might be summarized as an evolutionary or developmental approach to understanding the relationship between the three editions of the Guangshiyin yingyan ji 光世音應驗記 (or Guanshiyin yingyan ji 觀世音應驗記): (1) Guangshiyin yingyan ji 光世音應驗記, by Fu Liang 傅亮 (374–426), (2) Xu Guangshiyin yingyan ji 續光世音應驗記, by Zhang Yan 張演 in the mid-fifth cen., and (3) Xi Guanshiyin yingyan ji 繫觀世音應驗記, compiled by Lu Gao 陸杲 (459–532) in 501 and broader classifications of medieval Chinese literature, such as zhiguai 志怪 (accounts of the strange), yingyan 靈驗 (numinous or spiritual efficacy), and gantong 感通 (spiritual resonance).
Makita Tairyō published an edited and annotated edition of three of the earliest Chinese collections of miracle tales about Guanyin. Yü Chün-fang reports that he used a Japanese handcopied manuscript from the Kamakura period (1185–1333), which was preserved in Seirenji 西蓮寺in Kyoto, a Tendai temple, as the basis of this modern edition (Yü Chün-fang 2001, p. 158). The author, however, reports that they were discovered in the “Seiryo-in Temple” (l. 38). Please double check. Please confirm which temple is correct: Seirenji 西蓮寺or Seiryōin 清凉院?
The fundamental problem with this essay is not the content, it is the style and presentation. The biggest problem is that the paper is not consistent in the way that it refers to the numerous medieval texts and collections of zhiguai and other stories. It looks as though multiple people worked on translating the essay into English and paid no attention to correcting or settling on one standard form for referring to texts through romanized titles. Titles that appear repeatedly such as the Fayuan zhulin and Taiping guangji appear in numerous forms. The essay needs a complete vetting for style and grammar.
This reviewer has attached a pdf with many corrections and problems marked in green.
The title “The fluid Miraculous Stories” doesn’t really make much sense. Perhaps something like “The Evolution of Miraculous Stories of the Bodhisattva Avalokiteśvara in Medieval China: Centered on three editions of the Guanshiyin yingyan ji” would be better.
The author may need to explain why he/she uses the title Guangshiyin yingyan ji 光世音應驗記 instead of Guanshiyin yingyan ji 觀世音應驗記. Although it is true that Guangshiyin yingyan ji was the original title, both Makita Tairyō and Dong Qijiao use the title Guanshiyin yingyan ji in the titles of their books on these three texts—most likely because Guanshiyin became the standard name for the Bodhisattva Avalokiteśvara in the medieval period during the Tang dynasty.
In addition, when the author refers to stories in the three editions of this material, he/she adopts the citation style of providing the compiler’s last name, a dash, and the number of the story; for example, (Fu-5), seems to refer to the fifth story in Fu Liang’s Guangshiyin yingyan ji. Or or (Zhang - 7) [see l. 222], seems to refer to the seventh story in Zhang Yan’s Xu Guangshiyin yingyan ji. If this is correct, the author should explain this to the reader.
Foreign words, expressions, and book titles need to be italicized. In addition, the Sinographs (Chinese characters) for book titles or people’s names only need to be provided the first time they appear in the text. For the first half of the paper, the Sinographs for people’s names and book titles are provided nearly every time they appear. In addition, the author should use the Romanized
“Guangshiyin Yingyanji” and “Guangshiyin YingyanJi” or “YingyanJi” should be Guangshiyin yingyan ji.
“Ming Xiang Ji” 冥祥記 (l. 158) should be Mingxiang ji 冥祥記.
“Continued Biographies of Eminent Monks” (l. 342, 347) should be Xu gaoseng zhuan 續高僧傳.
“Book of Song” (l. 516) should be Song shu
If the author wants to provide an English translation, for the sake of clarity, it should be (History of the Liu-Song Dynasty)
“Biography of Famous Monks” (l. 586) should be Mingseng zhuan 名僧傳or Mingseng zhuan chao 名僧傳抄, depending on the author’s intent.
“Shamen Tansong Temple Record” (l. 623) should be Shamen tanzongsi ji 沙門曇宗寺記.
“Biography of the Lotus Sūtra” (l. 913, l. 1014) should be Fahua zhuanji 法華傳記 (Records of the Transmission of the Lotus Sūtra).
“Sheshi Yaolan” (l. 951) should be Shishi yaolan
“Fish Mountain and the Brahman Chant” (l. 960) should be provided in Chinese
“Miscellaneous Agama Sutra” (l. 1089) should be Za ahan jing 雜阿含經
“Zui Fu Baoying Sutra” (l. 1098) should be Foshuo zuifu baoying jing 佛說罪福報應經.
“Chu San Zang Ji Ji” (n. 26) should be Chu sanzang jiji.
These are just the most conspicuous examples. Many more could be shown.
“Shangyuan Jinlu Jianwen” 上元金箓簡文 (n. 26, p. 29) should be Shangyuan jinlu jianwen 上元金籙簡文 (Tablet of the Gold Register of the Superior Principle). Isn’t this a Taoist document? The traditional character 籙 should be used instead of 箓.
Regarding the “The Chapter of Guangming Anle Xing” 光明安樂行品 (l. 1031-1032, 1049, 1052, and 1060), the author should recognize that this term appears only in that story (see lines 1041-1046, although the similar expression “Chapter Guangming An Xing” 光明案行品 (l. 1074) appears in a different story (lines 1081-1086).
Fa Hu (l. 1050) should be Dharmarakṣa (Fahu 法護, ca. 223-300)
“The Chapter of An Xing” (l. 1050) should be the “Peaceful Practices” chapter
“The Chapter of Anle Xing” (l. 1051) should be “Comfortable Conduct” chapter
“The Chapter of Pu Mun” (l. 1051-1052) should be “The Gateway to Everywhere” chapter (Pumen pin), which is short for “The Gateway to Everywhere of the Bodhisattva Avalokiteśvara.” Although this is widely known, the author should acknowledge that the “Pumen pin” from Kumarajiva’s translation of the Lotus Sutra also circulated separately as the Guanshiyin jing (Guanyin Sutra).
What are the Sinographs for Pu Shou, which is translated as Mañjuśrī (l. 1059)? This reviewer thinks that this translation may be incorrect.
“Fayuan Zhulin·Guanyin Experience” (l. 586-587) is not completely clear and does not follow a standard citation style. Is Guanyin Experience a translation of “Ganying yuan” 感應緣?
If the author is using CBETA for the Taishō shinshū dai zōkyō 大正新修大藏經 [Taishō edition of the Buddhist canon], he/she should adopt a standard way of citation that includes the following information: Taishō number, volume number, page number, and register: for example, T 2122, 53.940a.
T = Taishō shinshū dai zōkyō 大正新修大藏經 [Taishō edition of the Buddhist canon]. Ed. Takakasu Junjirō 高楠順次郎, et al. 100 vols. Tokyo: Taishō Issaikyō Kankōkai, 1924–1935.
T 2122 = Fayuan zhulin 法苑珠林 [A forest of pearls in the garden of the Dharma]. 100 rolls. Compiled by Daoshi 道世 (ca. 596–683), completed in 668. T 2122, 53.269a–1030a.
For a Chinese audience or for an audience familiar with Chinese literature, the importance and authority of Lu Xun’s characterization of these texts as “Buddhist auxiliary texts” (l. 15-16) may be obvious. However, for a more general audience interested in Buddhism or Buddhist cultic practices, the significance and meaning of Lu Xun’s “Buddhist auxiliary texts” is not clear. Furthermore, the expression changdao 唱導 (l. 21) needs to be translated into English and explained. Does the author understand the term here as a noun, like changdaoshi 唱導師 (a preacher), or as a verb, “to preach to people and lead them to conversion”?
In addition, the author needs to explain his/her understanding and interpretation of zhiguai, yingyan, and gantong materials are. Although this reviewer understands, most readers do not have the background I possess. Furthermore, the author should be clear about when he thinks these different kinds of materials emerged in the medieval period. Although zhiguai are relatively well known in English secondary scholarship, yingyan and gantong are not. The words appear in titles of collections, but most scholars do not consider them distinctive “systems” of writing. For the most part, well-read readers see an evolutionary relationship between zhiguai and chuanqi 傳奇 (transmitted wonders) literature. In addition, the author should also explain the relationship between the terms gantong 感通 and ganying 感應 as he/she understands it. This explanation should appear somewhere between lines 874 and 888.
Several sources cited in the text are not found in the References section:
l. 43-44: Komina Ichiro (1982, p. 415-500)
l. 200: (Kominami Ichiro1982,p.418-434): Is this the same as above? Which is correct?
l. 430: “Yi Chu Liu Tie” (or Yichu liutie) is not in the primary sources section
l. 875-76: Shonan Ichiro
n. 4 (p. 27): Konan Yukinari’s research is not found in References
n. 52 (p. 31): Makoto Sano2020
n. 53 (p. 31): Robert Ford Company 2012
Other questions that need to be addressed:
l. 220: Is “supernatural fiction” the author’s gloss of zhiguai?
l. 224: Seng Rong (or Monk Rong) should be Sengrong
l. 774: What are “yingshi ji”?
The References need a complete vetting for style and transliteration. In addition, the two main sources for the stories are recorded wrongly:
Makita Tairyō 牧田諦亮, ed. Kanzeon ōkenki no kenkyū: Rikuchō koitsu 觀世音應驗記の硏究 : 六朝古逸 (Studies on the Tales of Guanshiyin’s Miraculous Manifestations: Lost texts surviving from the Six Dynasties). Kyoto: Heirakuji Shoten 平樂寺書店, 1970.
Dong Zhiqiao 董志翹, ed. “Guanshiyin yingyan ji sanzhong” yizhu 「觀世音應驗記三種」譯註 (Annotated translation of three editions of Tales of Guanshiyin’s Miraculous Manifestations). Nanjing: Jiangsu guji chubanshe 江蘇古籍出版社 and Jingxiao xinhua shudian 經銷新華書店, 2002.
The secondary sources are not presented in an appropriate manner. The author or translator has translated the titles when they should have been transliterated. For instance, the entry for Can Daoheng should be:
Cao Daoheng [provide sinographs]. 1992. “Lun Wang Yan huo tade ‘Mingxiang ji’ ” 论王琰和他的“冥祥记” (On Wang Yan and his Mingxiang ji). Wenxue yichan 文学遗产 (Literary Heritage) 1:26-36.
The entries for people with Western or European surnames are all done incorrectly.

Although the English is okay in many instances, the paper needs a full vetting because it is not consistent in how it presents the sources cited in the book. Please see the notes above.
Author Response
1.Please double check. Please confirm which temple is correct: Seirenji 西蓮寺or Seiryōin 清凉院?
Reply:Shōren-in 青蓮院
2.The title “The fluid Miraculous Stories” doesn’t really make much sense. Perhaps something like “The Evolution of Miraculous Stories of the Bodhisattva Avalokiteśvara in Medieval China: Centered on three editions of the Guanshiyin yingyan ji” would be better.
Reply:What I want to emphasize is the rewriting and generation of individual stories in different contexts. Many stories can be traced back to earlier prototypes found in Mingxiang Ji, Xuanyan Ji, Zhiguai, and even Xiaogan孝感 stories. Therefore, the Guanyin faith and Yingyan are just part of the transmission process of these stories. Nowtheless, the comment is very useful for me, so I change to the term "Miracle Stories in Motion" ,which might better to reflect the fluidity of the story content itself and the contextual rewriting within different systems.
3.The author may need to explain why he/she uses the title Guangshiyin yingyan ji 光世音應驗記 instead of Guanshiyin yingyan ji 觀世音應驗記. Although it is true that Guangshiyin yingyan ji was the original title, both Makita Tairyō and Dong Qijiao use the title Guanshiyin yingyan ji in the titles of their books on these three texts—most likely because Guanshiyin became the standard name for the Bodhisattva Avalokiteśvara in the medieval period during the Tang dynasty.
Reply:Based on the revised feedback, the usage conforms to Guanshiyin yingyan ji 觀世音應驗記
4,In addition, the author needs to explain his/her understanding and interpretation of zhiguai, yingyan, and gantong materials are. Although this reviewer understands, most readers do not have the background I possess. Furthermore, the author should be clear about when he thinks these different kinds of materials emerged in the medieval period. Although zhiguai are relatively well known in English secondary scholarship, yingyan and gantong are not. The words appear in titles of collections, but most scholars do not consider them distinctive “systems” of writing. For the most part, well-read readers see an evolutionary relationship between zhiguai and chuanqi 傳奇 (transmitted wonders) literature.
Reply:I have a explain on 574-578,881-888.And I have added further explain in the new version:
Previous research has focused on the evolutionary relationship between zhiguai (records of the strange) and chuanqi (transmitted wonders) literature. However, few scholars have noticed that these supernatural stories also have different writing systems depending on the authors' perspectives. Due to the different writing perspectives, we can distinguish Zhigui system written by non-Buddhist believers, Yingyansystem written by lay practitioners with Buddhist beliefs, and Gantongsystem written by Buddhist monks.
note58:The term of "Yingyan", originates from the Guangshiyin Yingyanji. Although the concept of Yingyan is not exclusive to lay practitioners. However, the concept of Yingyan refers to the purposefulness of their prayers and the immediacy of the response from the object of their faith. Therefore, choosing Yingyan represents the perspective of lay practitioners.
note59:The term of "Gantong", derived from the "Gantong"chapter in Xu gaoseng chuan, is intended to distinguish it from the system of supernatural and strange phenomena outside of Buddhist teachings.
5.Foreign words, expressions, and book titles need to be italicized. In addition, the Sinographs (Chinese characters) for book titles or people’s names only need to be provided the first time they appear in the text. For the first half of the paper, the Sinographs for people’s names and book titles are provided nearly every time they appear. In addition, the author should use the Romanized
Reply:corrected
6.Regarding the “The Chapter of Guangming Anle Xing” 光明安樂行品 (l. 1031-1032, 1049, 1052, and 1060), the author should recognize that this term appears only in that story (see lines 1041-1046, although the similar expression “Chapter Guangming An Xing” 光明案行品 (l. 1074) appears in a different story (lines 1081-1086).(譯本不同)
Reply:I have a brief explain in 1048-1056. But cancel the further explain because the essay is too long.
I believe that the term "案行品" (Anxing Pin) originated from the title "安行品" (Anxing Pin) in the Dharmaraksa version. The Song, Yuan, Ming, and Gong editions of Fayuan Zhulin all mention "安行品" (Anxing Pin), while only the Taishō Shinshū Dai Zōkyō uses "案行品" (Anxing Pin). Therefore, it is possible that the Taishō edition has a writing error. Additionally, both the "法苑珠林" (Fayuan Zhulin) and the "太平廣記" (Taiping Guangji) that adopt the term "安樂行品" (Anle Xing Pin) are derived from the Mingxiang Ji. Hence, it can be inferred that Mingxiang Ji adopted the translated name from the Dharmarakṣa version, while Yingyan Ji adopted the translated name from Kumārajīva.
7.For a Chinese audience or for an audience familiar with Chinese literature, the importance and authority of Lu Xun’s characterization of these texts as “Buddhist auxiliary texts” (l. 15-16) may be obvious. However, for a more general audience interested in Buddhism or Buddhist cultic practices, the significance and meaning of Lu Xun’s “Buddhist auxiliary texts” is not clear. Furthermore, the expression changdao 唱導 (l. 21) needs to be translated into English and explained. Does the author understand the term here as a noun, like changdaoshi 唱導師 (a preacher), or as a verb, “to preach to people and lead them to conversion”?
Reply:已修改,增加了註2:changdao唱導, refers to a preaching procedure in Buddhist rituals, where the speaker uses plain language and Vipāka or fateful stories to explain the principles and teachings found within Buddhist scriptures to the audience.
8.In addition, when the author refers to stories in the three editions of this material, he/she adopts the citation style of providing the compiler’s last name, a dash, and the number of the story; for example, (Fu-5), seems to refer to the fifth story in Fu Liang’s Guangshiyin yingyan ji. Or or (Zhang - 7) [see l. 222], seems to refer to the seventh story in Zhang Yan’s Xu Guangshiyin yingyan ji. If this is correct, the author should explain this to the reader.
Reply:The three editions of YingyanJi are referred to as the Fu edition, Zhang edition, and Lu edition by the author.And the number follow of the edition, refer to the number rank of the edition. For instance, Fu-7 refer to the seventh story in the Fu edition.
Reviewer 2 Report
The author's efforts can be seen. However, because of the length of the volume, there are many inconsistencies in notation and minor typographical errors, making it difficult to read. Often, it seems that references to literature are not properly made.
Pinyin notation of the first occurrence should be accompanied by Chinese characters.
eg. ll.65-66 Fu and Zhang editions and the later Lu Gao edition
l.43, Komina Ichiro, l.196, Kominami Ichiro > the same person?
Makita Taihō, Makita Teruaki, Makita Tairyō > the same person?
Shinohara Keiichi, Shinohara, Koichi > the same person?
Makoto Sano2020, Masako Sano > the same person? no data of Sano 2020 in the Secondary Source. cf. n.37.
l.222, Zhang - 7l. 362, Lu—63 etc.> no explanation of -7, -63 etc.
No need of indent for ll.621-626.
Shonan Ichiro > no ref. in the reference.
ll.855-872 > Fonts are different.
l.752 Zhiguai志怪, Yingyan應驗, and gantong感通 and l. 874, Zhiguai, Yingyan, and Kantong > referring to the same things?
l.874: 2.3 > 3.3
l.1088: The Five Realms Wheel五道輪, also known as五道輪.68 Describes>The Five Realms Wheel 五道輪, also known as 五道輪68 describes
Since this paper is not about reading Chinese texts, it is easier to read the original Chinese texts if they are included in the notes.
Normally, a paper does not leave a single line space when separating paragraphs. The authors themselves are aware that each chapter is long, which is probably why they take such a measure. Instead, why not separate each chapter with more subsections? Also, each subsection should clearly state where the problem lies and the conclusion, otherwise it is difficult to read.
It seems that the point of this paper exhausted in chapter 2, i.e., ll.538ff. and ll.981-989.
Chapter 3 is unusually long. At the very least, there should be a sentence at the end of chapter 2 that links to chapter 3.
The conclusion is also unusually long(ll.980-1114!) , and even cites the original Chinese texts in two places. The conclusion should only briefly describe the conclusions reached in the paper, and if there is anything further to be discussed, a fourth chapter should be created to discuss it. This is also related to what was pointed out earlier. If each chapter is divided into more subsections and the conclusion is clearly presented at the end of each section, the conclusion part can only briefly restate it. I believe that a major rewrite is needed.
The entire paper, including formatting, should be reworked by a native speaker.
Author Response
1.Chapter 3 is unusually long. At the very least, there should be a sentence at the end of chapter 2 that links to chapter 3.
Reply:I have added a sentence to link:Based on different writing perspectives and factors, these versions can be classified into different textual systems. The next chapter will focus on the study of the rewriting and generation of supernatural stories within different textual systems, examining the stories within the context of their overall transmission process. By exploring the continuous compilation and rewriting of supernatural stories, as well as the writing characteristics of different textual systems, we can better understand the fluidity of supernatural stories.
2.The conclusion is also unusually long(ll.980-1114!) , and even cites the original Chinese texts in two places. The conclusion should only briefly describe the conclusions reached in the paper, and if there is anything further to be discussed, a fourth chapter should be created to discuss it. This is also related to what was pointed out earlier. If each chapter is divided into more subsections and the conclusion is clearly presented at the end of each section, the conclusion part can only briefly restate it. I believe that a major rewrite is needed.
Reply:I have revised the conclusion to focus only on summarizing the content of the preceding chapters.
Round 2
Reviewer 1 Report
Although the author has made many corrections and provided a better title for the paper, this essay still suffers from many of the same problems that the first version it.
For a Chinese audience or for an audience familiar with Chinese literature, the importance and authority of Lu Xun’s characterization of these texts as “Buddhist auxiliary texts” (l. 15-16) may be obvious. However, for a more general audience interested in Buddhism or Buddhist cultic practices, the significance and meaning of Lu Xun’s “Buddhist auxiliary texts” is not clear. The author needs to explain the significance of the term “Buddhist auxiliary texts” and why he/she uses it. Does this have to do with the scholarly discourse on early Chinese literature in China? The author needs to explain to the audience why they should care.
Nowhere in the paper does the author explain or name the three editions of the Guangshiyin Yingyan ji, who authored them, and when they were written. The author needs to do this and explain that he/she will refer to (1) the Guangshiyin yingyan ji 光世音應驗記, by Fu Liang 傅亮 (374–426) as the Fu edition; (2) the Xu Guangshiyin yingyan ji 續光世音應驗記, by Zhang Yan 張演 in the mid-fifth cen., as the Zhang edition; and (3) the Xi Guanshiyin yingyan ji 繫觀世音應驗記, compiled by Lu Gao 陸杲 (459–532) in 501, as the Lu edition.
This should appear in the introductory section.
In addition, when the author refers to stories in the three editions of this material, he/she adopts the citation style of providing the compiler’s last name, a dash, and the number of the story; for example, (Fu-5), seems to refer to the fifth story in Fu Liang’s Guangshiyin yingyan ji. Or or (Zhang - 7) [see l. 221], seems to refer to the seventh story in Zhang Yan’s Xu Guangshiyin yingyan ji. The author must explain this to the reader.
In addition, the author discusses several classifications of medieval Chinese literature on lines 578 – 584., Readers should will be familiar with zhiguai 志怪 (accounts of the strange), but not yingyan 靈驗 (numinous or spiritual efficacy) and gantong 感通 (spiritual resonance). The author needs to explain his/her understanding and interpretation of zhiguai, yingyan, and gantong materials are. Although this reviewer understands, most readers do not have the background I possess. Furthermore, the author should be clear about when he thinks these different kinds of materials emerged in the medieval period. Although zhiguai are relatively well known in English secondary scholarship, yingyan and gantong are not. The words appear in titles of collections, but most scholars do not consider them distinctive “systems” of writing. For the most part, well-read readers see an evolutionary relationship between zhiguai and chuanqi 傳奇 (transmitted wonders) literature. In addition, the author should also explain the relationship between the terms gantong 感通 and ganying 感應 as he/she understands it.
The fundamental problem with this essay is not the content, it is still the style and presentation. The biggest problem is that the paper is still not consistent in the way that it refers to the numerous medieval texts and collections of zhiguai and other stories. It looks as though multiple people worked on translating the essay into English and paid no attention to correcting or settling on one standard form for referring to texts through romanized titles. Titles that appear repeatedly such as the Fayuan zhulin and Taiping guangji appear in numerous forms. The essay needs a complete vetting for style and grammar.
Foreign words, expressions, and book titles need to be italicized. In addition, the Sinographs (Chinese characters) for book titles or people’s names only need to be provided the first time they appear in the text. For the first half of the paper, the Sinographs for people’s names and book titles are provided nearly every time they appear. In addition, the author should use the Romanized version of book titles consistently. For instance, the author usually refers to the Mingxiangji as one word, but it also appears as Mingjiang ji (l. 756) and other ways. If the author prefers Mingxiangji as one word; then he/she should also consider following that style with other titles like Xuanyan ji (l. 400).
Although the body of the paper has been corrected somewhat, the endnotes and reference sections have not been corrected! The author must revise the endnotes and make corrections—and make sure that book titles are given the same way in Romanization all the time! Furthermore, many endnotes should be in-text citations.
The References section is also in terrible shape. The corrections suggested by me in the first version were for the most part not done. The titles are inconsistent with what is in the body of the paper and book titles are not italicized. Furthermore, the information for primary sources in the Taisho edition of the Buddhist Canon is not provided consistently.
Worse than the primary sources are the secondary sources! As I explained in the first version: The secondary sources are not presented in an appropriate manner. The author or translator has translated the titles when they should have been transliterated. For instance, the entry for Can Daoheng should be:
ALL OF THE SECONDARY SOURCES NEED TO BE CHECKED AND FIXED. Chinese and Japanese sources need to be in the following style so that interested readers can find these secondary sources:
Scholarly Article:
Cao Daoheng 曹道衡. 1992. “Lun Wang Yan huo tade ‘Mingxiang ji’ ” 论王琰和他的“冥祥记” (On Wang Yan and his Mingxiang ji). Wenxue yichan 文学遗产 (Literary Heritage) 1:26-36.
Scholarly Book:
Sano Seiko 佐野誠子. 2020. Kai o shirusu: Rikuchō shikai no tanjō to tenkai 怪を志す : 六朝志怪の誕生と展開 (Recording the Strange: The Birth and Development of Six Dynasties Zhiguai). Nagoya: Nagoya Daigaku Shuppankai 名古屋大学出版会.
The entries for people with Western or European surnames are all done incorrectly. In a bibliography, the last name is provided first, followed by a comma and the given name.
Knapp, Keith N.
Shinohara, Koichi and Jinhua Chen, eds.
Davis, Natalie Zemon
Company, Robert Ford.
THIS REVIEWER HAS ATTACHED ANOTHER CORRECTED FILE.

The fundamental problem with this essay is not the content, it is still the style and presentation. The biggest problem is that the paper is still not consistent in the way that it refers to the numerous medieval texts and collections of zhiguai and other stories. It looks as though multiple people worked on translating the essay into English and paid no attention to correcting or settling on one standard form for referring to texts through romanized titles. Titles that appear repeatedly such as the Fayuan zhulin and Taiping guangji appear in numerous forms. The essay needs a complete vetting for style and grammar.
Foreign words, expressions, and book titles need to be italicized. In addition, the Sinographs (Chinese characters) for book titles or people’s names only need to be provided the first time they appear in the text. For the first half of the paper, the Sinographs for people’s names and book titles are provided nearly every time they appear. In addition, the author should use the Romanized version of book titles consistently. For instance, the author usually refers to the Mingxiangji as one word, but it also appears as Mingjiang ji (l. 756) and other ways. If the author prefers Mingxiangji as one word; then he/she should also consider following that style with other titles like Xuanyan ji (l. 400).
Although the body of the paper has been corrected somewhat, the endnotes and reference sections have not been corrected! The author must revise the endnotes and make corrections—and make sure that book titles are given the same way in Romanization all the time! Furthermore, many endnotes should be in-text citations.
The References section is also in terrible shape. The corrections suggested by me in the first version were for the most part not done. The titles are inconsistent with what is in the body of the paper and book titles are not italicized. Furthermore, the information for primary sources in the Taisho edition of the Buddhist Canon is not provided consistently.
Worse than the primary sources are the secondary sources! As I explained in the first version: The secondary sources are not presented in an appropriate manner. The author or translator has translated the titles when they should have been transliterated. For instance, the entry for Can Daoheng should be:
ALL OF THE SECONDARY SOURCES NEED TO BE CHECKED AND FIXED. Chinese and Japanese sources need to be in the following style so that interested readers can find these secondary sources:
Scholarly Article:
Cao Daoheng 曹道衡. 1992. “Lun Wang Yan huo tade ‘Mingxiang ji’ ” 论王琰和他的“冥祥记” (On Wang Yan and his Mingxiang ji). Wenxue yichan 文学遗产 (Literary Heritage) 1:26-36.
Scholarly Book:
Sano Seiko 佐野誠子. 2020. Kai o shirusu: Rikuchō shikai no tanjō to tenkai 怪を志す : 六朝志怪の誕生と展開 (Recording the Strange: The Birth and Development of Six Dynasties Zhiguai). Nagoya: Nagoya Daigaku Shuppankai 名古屋大学出版会.
The entries for people with Western or European surnames are all done incorrectly. In a bibliography, the last name is provided first, followed by a comma and the given name.
Knapp, Keith N.
Shinohara, Koichi and Jinhua Chen, eds.
Davis, Natalie Zemon
Company, Robert Ford.
Author Response
1.For a Chinese audience or for an audience familiar with Chinese literature, the importance and authority of Lu Xun’s characterization of these texts as “Buddhist auxiliary texts” (l. 15-16) may be obvious. However, for a more general audience interested in Buddhism or Buddhist cultic practices, the significance and meaning of Lu Xun’s “Buddhist auxiliary texts” is not clear. The author needs to explain the significance of the term “Buddhist auxiliary texts” and why he/she uses it. Does this have to do with the scholarly discourse on early Chinese literature in China? The author needs to explain to the audience why they should care.
Reply: I had added the explain in note1:
The term "Buddhist auxiliary texts" was proposed by Lu Xun (1881-1936) in the early 20th century. Lu Xun is considered the founder of modern reserch of the ancient Chinese novel and one of the most important researchers in ancient Chinese literature. "Buddhist auxiliary texts" was the earlier definition for Buddhist miracle stories. It emphasizes the difference from normal Chinese novels and the purpose of Buddhist proselytizing. This concept was widely accepted by later researchers.
More broadly, Lu Xun defined "Buddhist auxiliary texts" as a kind of Zhiguai. Reconsidering the definition of "Buddhist auxiliary texts" in this approach helps us understand the complexity of Zhiguai and the development of medieval novels.
- Nowhere in the paper does the author explain or name the three editions of the Guangshiyin Yingyan ji, who authored them, and when they were written. The author needs to do this and explain that he/she will refer to (1) the Guangshiyin yingyan ji光世音應驗記, by Fu Liang 傅亮(374–426) as the Fu edition; (2) the Xu Guangshiyin yingyan ji 續光世音應驗記, by Zhang Yan 張演 in the mid-fifth cen., as the Zhang edition; and (3) the Xi Guanshiyin yingyan ji 繫觀世音應驗記, compiled by Lu Gao 陸杲 (459–532) in 501, as the Lu edition.
Reply: I had added the explain in (I:40-51):
Guangshiyin Yingyanji is a collective term for three different editions: 1. The first edition, known as the Guangshiyin Yingyanji (光世音應驗記), was written by Fu Liang 傅亮 (374–426) and is referred to as the Fu edition. 2. The second edition, called the Xu Guangshiyin Yingyanji 續光世音應驗記, was written by Zhang Yan 張演 in the mid-fifth century, and is referred to as the Zhang edition. 3. The third edition, named the Xi Guanshiyin Yingyanji 繫觀世音應驗記, was compiled by Lu Gao 陸杲 (459–532) in 501, and is known as the Lu edition.
Although these three editions were written by different authors, the later two editions mentioned the early edition and claimed to inherit its subject and compile it. Unfortunately, these three editions were lost in China after the Tang Dynasty, but they were rediscovered at the Shōren-in Temple (青蓮院) in Kyoto during the mid-20th century.
- In addition, when the author refers to stories in the three editions of this material, he/she adopts the citation style of providing the compiler’s last name, a dash, and the number of the story; for example, (Fu-5),seems to refer to the fifth story in Fu Liang’s Guangshiyin yingyan ji. Or or (Zhang - 7) [see l. 221], seems to refer to the seventh story in Zhang Yan’s Xu Guangshiyin yingyan ji.
Reply: I had a explanation in note17 and I add another explanation in note21.
note17:The three editions of Yingyanji are referred to as the Guanshiyin Yingyanji(Fu edition), Xu Guangshiyin Yingyanji(Zhang edition), and Xi Guanshiyin Yingyanji(Lu edition) by the author.And the number follow of the edition, refer to the number rank of the edition. For instance, Fu-5 refer to the fifth story in the Xi Guanshiyin Yingyanji(Fu edition) .
note21:The explainion form of this citation style is mentioned in Note16. (Zhang - 7) refer to the seventh story in Zhang Yan’s Xu Guangshiyin Yingyanji.
- In addition, the author discusses several classifications of medieval Chinese literature on lines 578 – 584., Readers should will be familiar with zhiguai志怪 (accounts of the strange), but not yingyan 靈驗 (numinous or spiritual efficacy) and gantong 感通 (spiritual resonance). The author needs to explain his/her understanding and interpretation of zhiguai, yingyan, and gantongmaterials are. Although this reviewer understands, most readers do not have the background I possess. Furthermore, the author should be clear about when he thinks these different kinds of materials emerged in the medieval period. Although zhiguai are relatively well known in English secondary scholarship, yingyan and gantong are not. The words appear in titles of collections, but most scholars do not consider them distinctive “systems” of writing. For the most part, well-read readers see an evolutionary relationship between zhiguai and chuanqi 傳奇 (transmitted wonders) literature. In addition, the author should also explain the relationship between the terms gantong 感通 and ganying 感應 as he/she understands it.
Reply: I move the note41(original note62) and note42(original note63) to the begin of the chapter 3. And I added the further explanation in (I:607-612) and note42
(I:607-612):Unlike the later-developed chuanqi, Yingyan, and Gantong in Zhiguai still retain descriptive traits and are mostly of short length. However, they exhibit different emphases when narrating the stories, leading to their selection and rewriting of the stories. As a result, the same story may take on various appearances and versions in the records of the three distinct narrative systems. Below, we will compare Yingyan with Zhiguai first.
note42(added):The reason why I prefer Gantong over Ganying (感應) is that Ganying is a term used in both Buddhist and non-Buddhist contexts, representing miraculous replies from any deity or god. On the other hand, Gantong is usually used to describe miraculous replies related to filial piety and the Buddha. Therefore, Gantong is a more suitable term to represent the stories written by monks, reaffirming the close relationship between filial piety stories and Buddhist stories.
Reviewer 2 Report
Content is good. The composition has improved to some extent. However, it is not in a state to be published unless the many remaining typographical errors are corrected. The authors are strongly encouraged also to ask a few colleagues who are good at writing in English to spend at least one week carefully correcting typographical errors and formatting.
A few examples.
Nabhatra 求那跋陀羅. Cf. Qiunabatuoluo 求那跋陀羅 in T 263. (Another reviewer should have already pointed this out in the first peer review)
Sano Masako>Sano Seiko
Note 57: Seiko Sano>Sano Seiko
Masashii>Masashi
Note 5: Liu Yeqiu, 1987, p. 83Kyoto:Kyoto: >?
Terutaka Kawamura河村孝照>Kawamura Kōshō 河村孝照
There should be a space after , and .
佛所行贊[Praise of the Buddha]>[Praise of the Buddha's Acts]
Author Response
I have corrected the typographical errors and added some further explanation on some key concept.
Round 3
Reviewer 1 Report
This revised paper has addressed many of the issues this reviewer has been concerned about. This reviewer has attached another corrected version of the paper. The body of the paper only has a few typographical errors and stylistic errors.
The errors that remain are primarily in the references section. The author needs to make sure he/she has the proper diacritical marks when transliterating titles from Japanese.
The transliterations from Chinese have been omitted in some cases and are typically sloppy. The author needs to understand the conventions.

Author Response
I have asked a native speaker to revise and correct the references section, as well as transliterate titles from Japanese.
Reviewer 2 Report
There is no problem with the content, but the author's indifference to formatting and consistency of notation is appalling. One place that another reviewer pointed out at the first review and that I pointed out at the second review (Nabhatra) has not been fixed, and one place that I pointed out in the second review (Kawamura) has not been fixed either.
Once again, the authors should have colleagues familiar with writing English papers or native speakers of English thoroughly examine the text and formatting and then sincerely correct any typographical errors.
I also suggest that the editors of Religions give authors a week or more, rather than a few days, to make corrections. Such a number of typographical errors is a severe problem that also affects the journal's credibility.
Revise as appropriate according to the following instructions. Not all are pointed out. Note that italics are not reflected for the convenience of writing here. Line numbers are not noted, but should be able to be found by searching the file.
No extra space is needed at the beginning of the quotation mark.
" Xu>"Xu
Unify with or without a space after AD.
AD414>AD 414
Lu—63>Lu—63
Zhang - 7
Fu-5
Italicize Lu, Zhang, and Fu.
Wang Xin, 2018, p. 128-140.Wei Bin, 2012, pp. 39-48.Wang Xin, 2017, pp. 137-145
>
Wang Xin, 2018, pp. 128-140. Wei Bin, 2012, pp. 39-48. Wang Xin, 2017, pp. 137-145.
Use pp. to refer to multiple pages. Insert a space after the ".". Modify others accordingly.
Sakano Masako (2020, p. 238-267).Yu Junfang (2002,p. 167-182).
>
Sano Seiko (2020, pp. 238-267).Yu Junfang (2002, pp. 167-182).
In addition to the above, insert a space after the ",". Other items should also be modified accordingly.
According to Seiko Sano’s research>According to Sano Seiko’s research
Sano Seiko's book "Striving for the Extraordinary: The Birth and Development of Six Dynasties Tales of the Strange"
>
Sano Seiko's book "Recording the Strange: The Birth and Development of Six Dynasties Zhiguai"
As another reviewer pointed out, 志す means Recording, not Striving for. This misreading makes this reviewer extremely uneasy not only about the author's Japanese reading ability but also about the quality of the main thesis, since 志怪 is exactly the Zhiguai, the subject of this paper.
Who or what is the "Nankai" in "Nankai's research" in note 18? No mentioning elsewhere.
T 417,=Banzhou Sanmei Jing 般舟三昧經[pratyutpannasamadhi sutra] Comp. by Jilou Archen 支婁迦讖.T 417, 13.
>
T 417=Banzhou Sanmei Jing 般舟三昧經 [Pratyutpannasamadhi sutra] Trans. by Zhiloujiachen 支婁迦讖. T 417, 13.
Put a space before [ and (.
T0747a=Fu shuo zui fu baoying jing 佛說罪福報應經[The Sutra of Karmic Retribution as Taught by the Buddha] Comp. by Nabhatra求那跋陀羅. T 0747a, 17.
Why a? Not Comp but Trans. Not Nabatra as is pointed out by reviewers twice.
>
T747=Fu shuo zui fu baoying jing 佛說罪福報應經 [The Sutra of Karmic Retribution as Taught by the Buddha] Trans. by Qiunabatuoluo 求那跋陀羅. T 747, 17.
By the way, neither this T 747 nor T 99 seems to have any mention in the text or notes. Please examine the other references carefully and remove any references that are not mentioned in the text or notes.
Nakajima Nagafum, Ito Reiko, Hirata Masashi>Nakajima Osafumi, Ito Reiko, Hirata Shoji
Takakasu>Takakusu
Terutaka Kawamura>Kawamura Kōshō
I strongly recommend that the author have a native speaker revise the entire document for formatting as well as English.
Author Response
I have asked a native speaker to revise and correct the references section and entire document again, as well as transliterate titles from Japanese.
Round 4
Reviewer 2 Report
Correct T 1311 to T 1331. Reconsider the English translation of 佛說灌頂經. While there are still a few flaws, most of the other formatting flaws seem to have been corrected. I leave the rest to the editors.
nothing in particular
Author Response
Correct T 1311 to T 1331. Change the translation of 佛說灌頂經 to
Consecration Sutra.